# Increased levels of mitochondrial import factor Mia40 prevent the aggregation of polyQ proteins in the cytosol

Anna M Schlagowski[1], Katharina Knöringer[1] (iD), Sandrine Morlot[2,3,4,5], Ana Sánchez Vicente[6], Tamara Flohr[1] (iD), Lena Krämer[1], Felix Boos[1] (iD), Nabeel Khalid[7], Sheraz Ahmed[7], Jana Schramm[8], Lena M Murschall[9], Per Haberkant[10], Frank Stein[10] (iD), Jan Riemer[9] (iD), Benedikt Westermann[8] (iD), Ralf J Braun[8,11] (iD), Konstanze F Winklhofer[6] (iD), Gilles Charvin[2,3,4,5] (iD) & Johannes M Herrmann[1,*] (iD)

## Abstract

The formation of protein aggregates is a hallmark of neurodegenerative diseases. Observations on patient samples and model systems demonstrated links between aggregate formation and declining mitochondrial functionality, but causalities remain unclear. We used *Saccharomyces cerevisiae* to analyze how mitochondrial processes regulate the behavior of aggregation-prone polyQ protein derived from human huntingtin. Expression of Q97-GFP rapidly led to insoluble cytosolic aggregates and cell death. Although aggregation impaired mitochondrial respiration only slightly, it considerably interfered with the import of mitochondrial precursor proteins. Mutants in the import component Mia40 were hypersensitive to Q97-GFP, whereas Mia40 overexpression strongly suppressed the formation of toxic Q97-GFP aggregates both in yeast and in human cells. Based on these observations, we propose that the post-translational import of mitochondrial precursor proteins into mitochondria competes with aggregation-prone cytosolic proteins for chaperones and proteasome capacity. Mia40 regulates this competition as it has a rate-limiting role in mitochondrial protein import. Therefore, Mia40 is a dynamic regulator in mitochondrial biogenesis that can be exploited to stabilize cytosolic proteostasis.

**Keywords** huntingtin; Mia40; mitochondria; protein aggregation; protein translocation
**Subject Categories** Membranes & Trafficking; Neuroscience; Translation & Protein Quality

**The EMBO Journal (2021) 40: e107913**
See also: **U Nowicka et al** (August 2021)

## Introduction

The accumulation of misfolded or aggregated proteins is frequently observed in stressed cells. Increased levels of misfolded proteins can be the cause, but also the consequence of disease states (for overview, see Chiti & Dobson, 2017; Sontag, Samant *et al*, 2017; Vaquer-Alicea & Diamond, 2019). Owing to their capability to sequester chaperones and to occupy proteases, misfolded proteins can route metastable or slowly folding polypeptides into aggregates (Gidalevitz, Ben-Zvi *et al*, 2006; Kim, Hosp *et al*, 2016). This might induce a calamitous amplifying reaction which finally leads to proteotoxicity and cell death.

Cells employ a number of different strategies to avoid the hazardous accumulation of misfolded proteins, including their stabilization and disaggregation by chaperones, their degradation by proteases, and their controlled sequestration into aggregates. Mitochondrial functionality modulates cytosolic protein homeostasis in several ways: (i) Protein aggregates often associate with the mitochondrial outer membrane (Aguilaniu, Gustafsson *et al*, 2003; Liu, Lillo *et al*, 2004; Ferri, Cozzolino *et al*, 2006; Gruber, Hornburg *et al*, 2018; Suhm, Kaimal *et al*, 2018; Yablonska, Ganesan *et al*, 2019); in yeast, this property is used to retain misfolded proteins in

1 Cell Biology, University of Kaiserslautern, Kaiserslautern, Germany
2 Institut de Génétique et de Biologie Moléculaire et Cellulaire, Illkirch, France
3 Centre National de la Recherche Scientifique, UMR7104, Illkirch, France
4 Institut National de la Santé et de la Recherche Médicale, U964, Illkirch, France
5 Université de Strasbourg, Illkirch, France
6 Department of Molecular Cell Biology, Institute of Biochemistry and Pathobiochemistry, Ruhr University Bochum, Bochum, Germany
7 German Research Center for Artificial Intelligence DFKI, Kaiserslautern, Germany
8 Cell Biology, University of Bayreuth, Bayreuth, Germany
9 Biochemistry, University of Cologne, Cologne, Germany
10 Proteomics Core Facility, EMBL Heidelberg, Heidelberg, Germany
11 Neurodegeneration, Danube Private University, Krems/Donau, Austria
*Corresponding author. Tel: +49 631 2052406; Fax: +49 631 2052492; E-mail: hannes.herrmann@biologie.uni-kl.de

the mother in order to prevent their inheritance to daughter cells (Zhou, Slaughter *et al*, 2011; Mogk & Bukau, 2014; Böckler, Chelius *et al*, 2017). (ii) Cytosolic aggregates impair the import of precursor proteins from the cytosol into mitochondria (Li, Vande Velde *et al*, 2010; Napoli, Wong *et al*, 2013; Yano, Baranov *et al*, 2014; Cenini, Rub *et al*, 2016). (iii) *Vice versa*, mitochondrial precursor proteins in the cytosol challenge proteostasis and sequester chaperones and the ubiquitin–proteasome system (Wrobel, Topf *et al*, 2015; Boos, Krämer *et al*, 2019; Poveda-Huertes, Matic *et al*, 2020). In this context, the hydrophobic carrier proteins of the inner membrane appear to be of specific relevance owing to their hydrophobic nature and specific import mechanism (Williams, Jan *et al*, 2014; Wang & Chen, 2015). (iv) Modulating mitochondrial activities can induce stress resistance pathways and thereby mitigate the accumulation and toxicity of cytosolic aggregates (Mason, Casu *et al*, 2013; Labbadia, Brielmann *et al*, 2017; Sorrentino, Romani *et al*, 2017; Fessler, Eckl *et al*, 2020; Guo, Aviles *et al*, 2020; Straub, Weraarpachai *et al*, 2021). (v) It was proposed that aggregated cytosolic proteins are disentangled on the mitochondrial surface and imported into the mitochondria to enable their degradation by mitochondrial proteases (Ruan, Zhou *et al*, 2017; Li, Xue *et al*, 2019) or removal by mitophagy (Guo, Ma *et al*, 2012; Hwang, Disatnik *et al*, 2015; Khalil *et al*, 2015).

The implications of mitochondrial biology for cytosolic proteostasis are even more complicated due to the central role of mitochondria in energy metabolism and redox homeostasis; apparently, the pathways relevant for cytosolic aggregate formation and mitochondrial functionality are intertwined in many ways. Yeast cells have been extensively used in the past as a rather simple and well-defined model system to unravel details of cytosolic homeostasis with a special focus on the role of mitochondrial protein import for the formation of cytosolic aggregates (Lu, Psakhye *et al*, 2014; Yang, Hao *et al*, 2016; Gruber *et al*, 2018; Weidberg & Amon, 2018). In order to study the role of mitochondrial protein biogenesis for cytosolic proteostasis, we expressed an established aggregating model protein in the baker's yeast *Saccharomyces cerevisiae*. The first exon of human huntingtin encodes an aggregation-prone poly-glutamine stretch (polyQ). Expression of polyQ fragments in yeast faithfully recapitulates huntingtin aggregation in a polyQ length-dependent manner (Braun, Buttner *et al*, 2010). We observed that the expression levels, mitochondrial localization, and functionality of Mia40 are critical determinants for polyQ toxicity. Mia40 (in humans also CHCHD4) is the rate-limiting essential factor of the machinery that imports proteins into the intermembrane space (IMS) of mitochondria (Chacinska, Pfannschmidt *et al*, 2004; Naoe, Ohwa *et al*, 2004; Peleh, Cordat *et al*, 2016; Peleh, Zannini *et al*, 2017; Habich, Salscheider *et al*, 2019). Increased levels of Mia40 in the IMS counteract the occurrence of aggregate-inducing nucleation seeds formed by the prion-like protein Rnq1 and suppress the growth arrest induced by an aggregation-prone polyQ protein. Overexpression of other components of the mitochondrial import machinery, in particular those relevant for the biogenesis of membrane proteins, was also able to suppress polyQ toxicity, pointing to a particular relevance of hydrophobic precursor proteins for cytosolic proteostasis. In general, our results show that the regulation of the mitochondrial import machinery, and in particular the modulation of the levels of Mia40, serves as an efficient molecular mechanism to finetune cytosolic protein homeostasis.

# Results

## Expression of Q97-GFP rapidly induces cytosolic aggregates and stalls cell growth

The first exon of human huntingtin, comprising the polyQ stretch fused to green fluorescent protein (GFP), has been used in the past to study the formation of aggregates in the yeast cytosol (Krobitsch & Lindquist, 2000; Duennwald, Jagadish *et al*, 2006; Klaips, Gropp *et al*, 2020). We expressed a non-aggregating variant of 25 and an aggregation-prone variant of 97 glutamine residues fused to GFP (Q25-GFP and Q97-GFP, Fig 1A) from a regulatable *GAL1* (for short here also referred as *GAL*) promoter in wild-type yeast cells. We initially cultured the cells in a galactose-free lactate medium to induce respiration and to stimulate mitochondrial biogenesis, before we induced the expression of the polyQ proteins by shifting cells to lactate medium that contained 2% galactose. Upon induction in galactose-containing medium, Q25-GFP expression resulted in a homogeneous cytosolic distribution, whereas Q97-GFP was predominantly forming aggregates seen as punctate signals (Fig 1B). Expression of Q97-GFP but not that of Q25-GFP was toxic and prevented cell growth (Fig 1C).

To analyze the order of events in polyQ-mediated toxicity, we analyzed cells after different times of induction. The Q97-GFP mRNA was detectable already after 10 min of induction and reached a maximum after about 90 min (Fig 1D). The Q97-GFP protein was well observed after 30 min and also reached a maximum after 90–120 min (Fig 1E). Interestingly, initially the Q97-GFP protein gave rise to a 47 kDa band on SDS gels closely matching the calculated molecular weight of the protein of 42.9 kDa. However, at later time points most of the protein was detected at the upper edge of the gel, indicative for the formation of SDS-insoluble protein aggregates (Douglas, Summers *et al*, 2009; Kim *et al*, 2016). The amounts of aggregated protein are presumably underestimated due to the limited solubility of this protein species.

This aggregation behavior was confirmed by fluorescence microscopy where an initially homogeneous GFP signal was replaced by aggregates several hours after induction in an increasing number of cells (Fig 1F). After 4 h, about 50% of all cells that showed green fluorescence contained aggregates which increased to almost 100% after 6 h of expression (Fig 1G). Thus, the Q97-GFP protein is initially soluble and homogeneously distributed in the cytosol, but then rapidly aggregates in basically all cells, which coincides with a growth arrest.

## A functionally compromised mutant of the mitochondrial import factor Mia40 shows increased Q97 toxicity

Previous studies suggested that cytosolic protein aggregation can disturb the functionality of mitochondria (Solans, Zambrano *et al*, 2006; Mossmann, Vogtle *et al*, 2014; Papsdorf, Kaiser *et al*, 2015). We therefore tested whether the Q97-GFP aggregates co-localize with mitochondria (Fig 2A). We observed that mitochondria and aggregates were clearly distinct structures. Mitochondrial co-localization that was seen for a few puncta might represent random contacts. Moreover, the formation of the Q97-GFP aggregates did not destroy the mitochondrial network which maintained its reticular structure for at least 4 h after induction of Q97-GFP expression

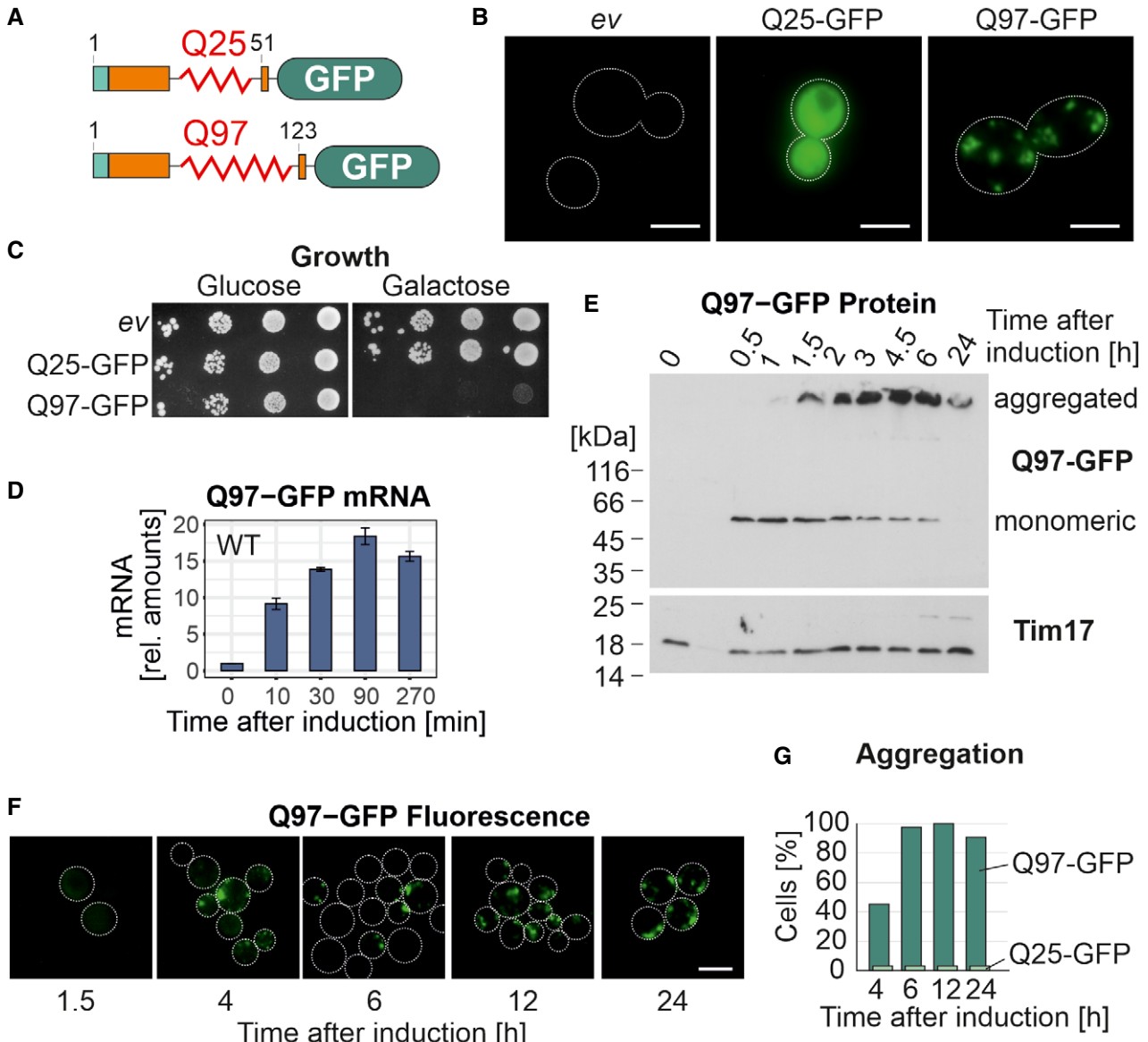

**Figure 1. The expression of Q97-GFP in the cytosol leads to the formation of insoluble protein aggregates.**

A  Schematic representation of the structure of the Q25-GFP and Q97-GFP model proteins used in this study. The sequence consists of a FLAG-tag (turquois) followed by the polyQ domain from huntingtin exon1 (orange) and eGFP.

B  Q25-GFP and Q97-GFP were expressed in wild-type cells for 18 h before cells were visualized by widefield fluorescence microscopy. Cells harboring an empty vector (*ev*) are shown for control. Cell boundaries are indicated by dashed lines. Bars, 5 μm.

C  Wild-type cells expressing the indicated proteins under control of a galactose-inducible *GAL1* promoter were grown in glucose medium to mid-log phase before tenfold serial dilutions were dropped on plates containing glucose or galactose.

D  Wild-type (WT) cells were shifted from glucose to galactose. Then, mRNA levels of Q97-GFP were measured by qPCR. Shown are normalized mean values and standard deviations from three replicates.

E  Cells were lysed with SDS-containing sample buffer before proteins were visualized by Western blotting. Insoluble aggregates migrate at the top of the gel between stacker and resolving gel. Tim17 served as loading control.

F  Microscopic images of the Q97-GFP fluorescence. Bars, 5 μm.

G  The percentage of cells containing or lacking detectable aggregates was quantified. Cell boundaries are indicated by dashed lines.

Source data are available online for this figure.

(Fig 2A). We also monitored the respiration-driven oxygen consumption and the activity of cytochrome *c* oxidase which was only moderately reduced, even after 24 h of Q97-GFP expression

(Fig 2B and C). We conclude that the expression of Q97-GFP induces the formation of cytosolic aggregates and impairs cell growth; however, at least within the first hours of induction, it only

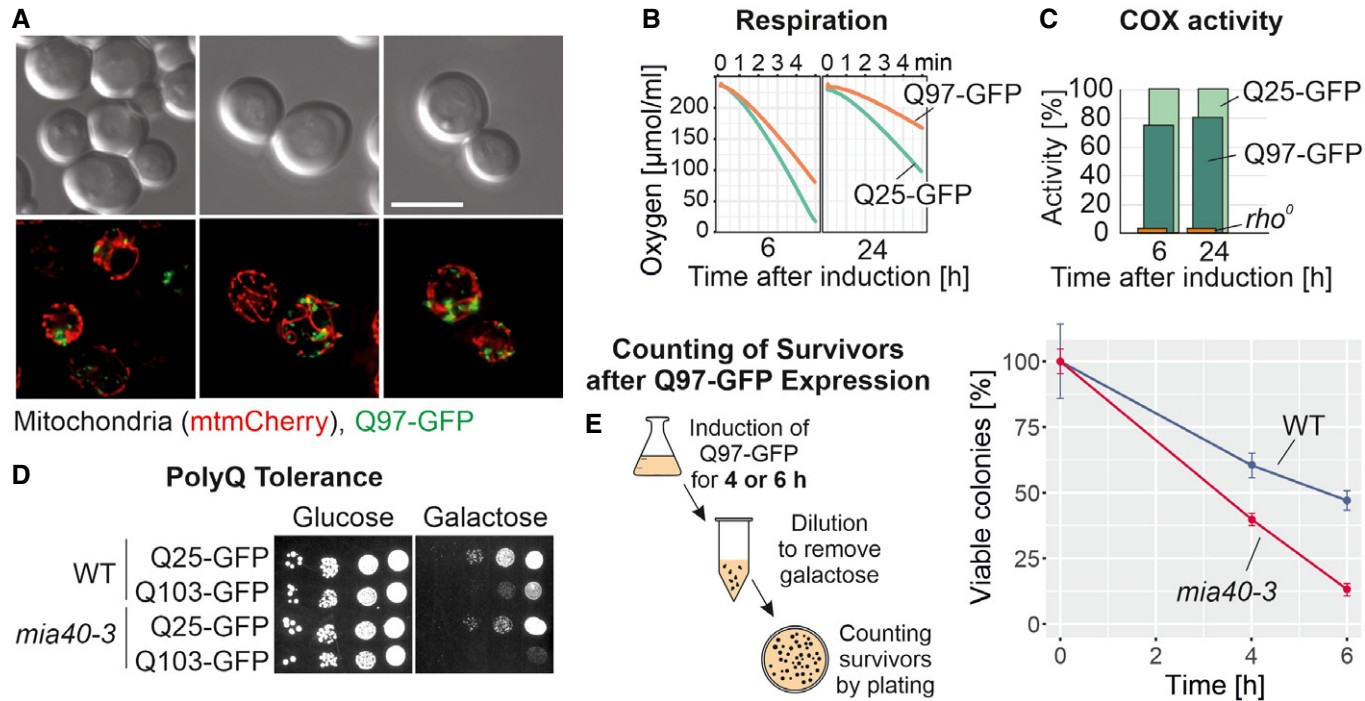

**Figure 2. Temperature-sensitive *mia40-3* cells are hypersensitive to polyQ aggregates.**

A    Mitochondria were visualized by mitochondrially targeted mCherry (red) in WT cells expressing Q97-GFP for 4 h. Fluorescence micrographs are maximum intensity projections of z stacks subjected to deconvolution. Bar, 5 μm.

B, C    The polyQ proteins were expressed for the indicated time periods. Mitochondria were isolated from wild-type strains containing the Q25/97-GFP-expressing plasmids and from cells lacking mitochondrial DNA (*rho⁰*). The ability of these mitochondria to respire (shown as NADH-induced oxygen consumption) or their activity of cytochrome *c* oxidase (shown as their capacity to oxidize reduced cytochrome *c*) was measured.

D    Cells of the temperature-sensitive *mia40-3* mutant (Chacinska *et al*, 2004) or the corresponding wild type were transformed with plasmids expressing the indicated proteins from a low-expression *GALL* promoter (Mason *et al*, 2013) and grown on the respective carbon sources at permissive conditions (25°C).

E    Q97-GFP was expressed in *mia40-3* and corresponding wild-type cells for the times indicated. Cells were harvested and washed, and survivors were counted after plating on glucose plates. Mean values and standard deviations of three replicates are shown.

Source data are available online for this figure.

---

has mild effects on mitochondrial morphology and the functionality of the respiratory chain.

Next, we tested whether mitochondrial functionality influenced polyQ toxicity. Since the expression of the Q97-GFP from the *GAL* promoter was highly toxic, we used a plasmid which expressed Q103-GFP from a low-expression *GALL* promoter (Mason *et al*, 2013) and tested its effect on the growth in a temperature-sensitive Mia40 (*mia40-3*) mutant. Mia40 is an essential protein of the mitochondrial protein import machinery (Chacinska *et al*, 2004; Naoe *et al*, 2004), and the discovery that cytosolic precursors of mitochondrial proteins are toxic was initially made in such a temperature-sensitive Mia40 mutant (Wrobel *et al*, 2015). We observed that the *mia40-3* mutant was more sensitive to Q103-GFP expression than the wild type: Even at permissive temperatures, cells were unable to grow on galactose-containing plates (Fig 2D). This suggests that the burden of non-imported mitochondrial precursors in the cytosol adds to the problems caused by toxic polyQ proteins.

To test whether the effects on growth are due to a growth arrest or to cell death, we expressed high levels of Q97-GFP in wild-type and *mia40-3* cells and grew them to log phase. Cells were exposed for 4 or 6 h to galactose, reisolated, and washed before surviving cells were counted by a plating assay on glucose medium. We observed that viability of Q97-GFP-expressing cells rapidly declined, in particular in the *mia40-3* mutant (Fig 2E), suggesting that the growth arrest is caused by cell death (Chacinska, Lind *et al*, 2005).

**Overexpression of Mia40 suppresses the aggregation and toxicity of Q97-GFP**

The observed hypersensitivity of *mia40-3* cells to Q97-GFP expression inspired us to test the effect of Mia40 overexpression. To this end, we made use of a strain which expresses Mia40 from a *GAL* promoter and which, in addition, harbors a plasmid with *MIA40* under the control of its endogenous promoter (Terziyska, Grumbt *et al*, 2007). This extra-copy of *MIA40* allows the strain to grow on glucose where the *GAL*-driven expression is repressed. This strain was transformed with the Q25-GFP and Q97-GFP expression plasmids. To our surprise, we observed that co-overexpression of Mia40 indeed strongly protected cells against the Q97-GFP-mediated growth arrest (Fig 3A). The Mia40-induced suppression was not due to reduced Q97-GFP expression as *GAL*-Mia40 cells even contained

higher Q97-GFP levels than wild-type cells, albeit the levels of Q97-GFP were much lower than those of the non-toxic Q25-GFP variant (Fig 3B). Intriguingly, upon Mia40 overexpression Q97-GFP hardly formed any SDS-resistant aggregates, indicating that elevated levels of Mia40 suppressed aggregation of Q97-GFP.

Fluorescence microscopy confirmed the striking difference in aggregate formation (Fig 3C and D): Whereas Q97-GFP formed several small, distinct aggregates in wild-type cells, the protein was either homogeneously dispersed in the *GAL*-Mia40 overexpression strain or formed one large intracellular aggregate. Such large polyQ aggregates were described before in mutants of cytosolic chaperones (Krobitsch & Lindquist, 2000; Meriin, Zhang *et al*, 2002; Dehay & Bertolotti, 2006; Yang *et al*, 2016; Higgins, Kabbaj *et al*, 2018).

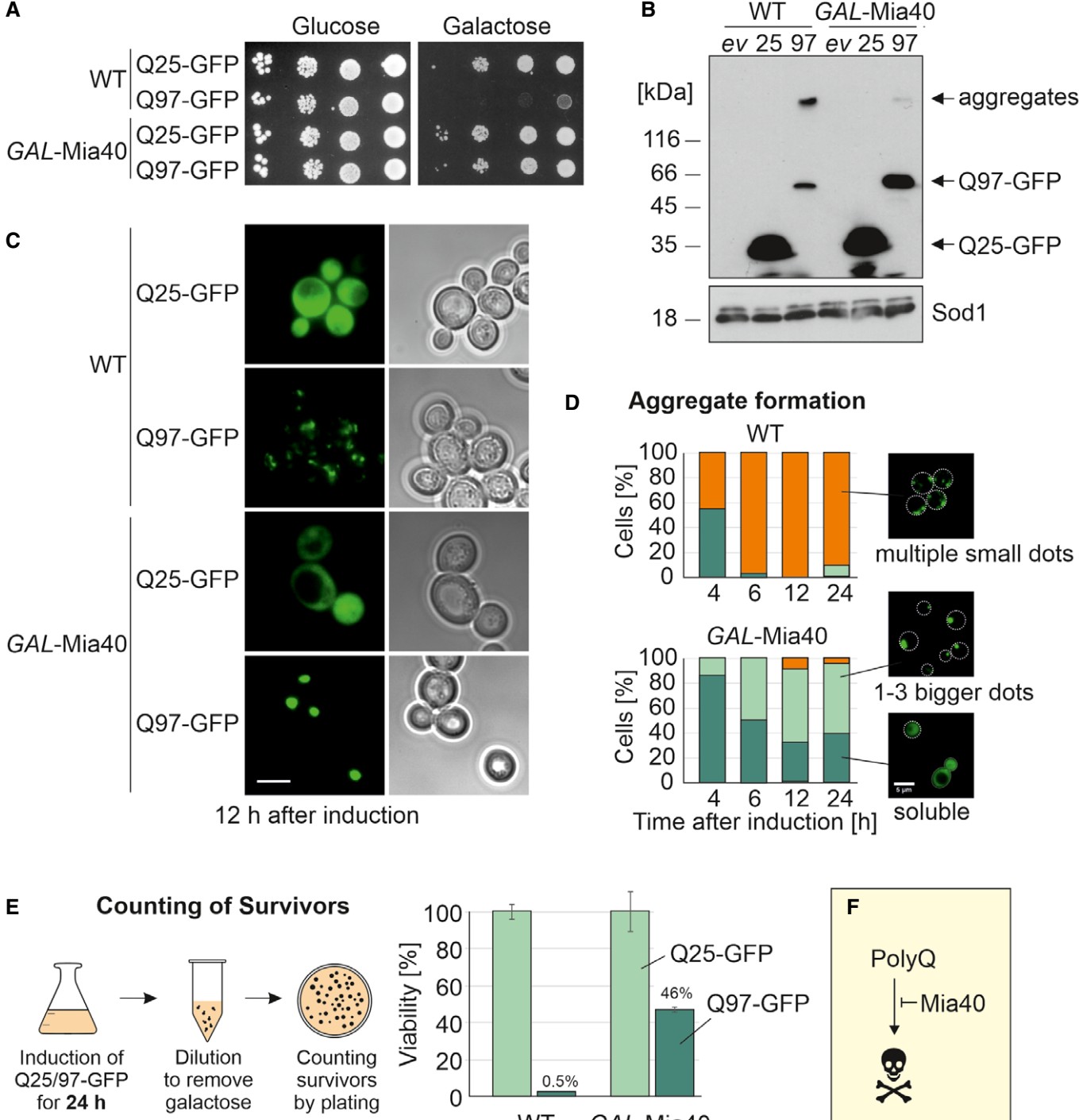

**Figure 3.**

**Figure 3.  Overexpression of Mia40 suppresses polyQ toxicity in yeast.**

A   Indicated strains were grown on glucose medium before tenfold serial dilutions were dropped on glucose- or galactose-containing plates.
B   Cell extracts were analyzed by Western blotting after shifting cultures to galactose for 16 h. 25, Q25-GFP; 97, Q97-GFP; and *ev*, empty vector.
C   Microscopy images of the indicated strains 12 h after shifting them to galactose. Note that in *GAL*-Mia40 cells the form and number of aggregates is very different to WT cells. Bar, 5 μm.
D   The patterns of the Q97-GFP distribution were quantified after different time of expression, *n* = 100. Cell boundaries are indicated by dashed lines.
E   PolyQ proteins were expressed for 24 h before survivors were counted. Mean values and standard deviations from three independent experiments are shown.
F   The *GAL*-Mia40 strain was much more resistant to Q97-GFP expression than WT cells, indicating that high Mia40 levels can suppress polyQ toxicity.

Source data are available online for this figure.

Upon expression of Q25-GFP or Q97-GFP, mitochondria still formed a wild type-like reticulate structure, both in the presence and in the absence of Mia40 overexpression (Fig EV1). In some cells that simultaneously overexpressed Q97-GFP and Mia40, we noticed more reticulate, "curly" mitochondria (Fig EV1). These structures were not observed when only one of these proteins was overexpressed. We are not aware that such structures were reported before, and their significance is currently unclear.

Mia40 overexpression was very effective in the repression of polyQ toxicity and about half of all cells even survived the *GAL*-driven expression of Q97-GFP for 24 h (Fig 3E). From this, we conclude that the overexpression of the mitochondrial import factor Mia40 can protect against the formation of toxic polyQ aggregates (Fig 3F).

## Mia40 influences the formation and inheritance of polyQ aggregates

Since we had observed that induction of Q97-GFP in wild-type cells resulted in an initially homogeneously distributed protein that later formed SDS-resistant aggregates, we wanted to better understand the temporal and spatial patterns of aggregate formation and inheritance in wild-type and *GAL*-Mia40 cells. To this end, we monitored the growth of wild-type and *GAL*-Mia40 cells after galactose-driven induction of Q25-GFP and Q97-GFP in a microfluidics growth chamber (Morlot, Song *et al*, 2019) under the fluorescence microscope (Fig 4A). Thereby, individual cells could be followed over time which showed that *GAL*-Q25 levels did not interfere with bud formation and growth. In contrast, in wild-type cells Q97-GFP expression slowed down cell division, and cells died prematurely (Fig 4B, Movies EV1–EV3). The pattern was entirely different in *GAL*-Mia40 cells: Cells accumulated Q97-GFP with a homogeneous intracellular distribution and thereby reached higher fluorescence levels than wild-type cells (Fig 4B and C). However, we noticed that, in individual cells, occasionally all the evenly distributed fluorescence collapsed indicating the formation of one characteristic large aggregate (Fig 4B). This collapse was very sudden and occurred from one frame to the next, i.e., in less than 10 min and without changing the total fluorescence intensity of a cell (Fig 4D). There was no noticeable correlation between the level of Q97-GFP signal and the time point of the collapse, suggesting that the formation of these large aggregates was a stochastic process. Once cells had these large aggregates formed, they ceased to produce new buds. Obviously, they were arrested in cell cycle and died after some time (Movie EV4). Even when galactose was removed to stop Q97-GFP expression, these aggregates persisted, whereas the diffuse Q97-GFP was degraded (Fig 4E).

To better follow the distribution of the GFP patterns in these cells, we developed an artificial intelligence tool which was trained to recognize individual yeast cells and the different patterns within these cells (i.e., unstained, homogeneous, smaller aggregates, one large aggregate) (Fig EV2A). This algorithm quantified in an unbiased manner the different categories of cells in the four strains (Fig EV2B). This allowed us to distinguish three distinct scenarios (Fig 4F): (i) Soluble Q25-GFP homogeneously accumulates in all cells over time. (ii) Q97-GFP forms small aggregates in basically all wild-type cells, slows down growth, and leads to premature cell death. (iii) In *GAL*-Mia40 cells, several different patterns emerged: Some cells formed the large aggregates which stopped their growth, whereas other cells continued to grow and accumulated homogeneously distributed Q97-GFP. Thus, the large aggregates of the *GAL*-Mia40 cells are not better tolerated than the aggregates in the wild type, but seem to be even more toxic. However, since they occur only sporadically, cells forming these structures are outnumbered by well growing cells showing homogeneous Q97-GFP staining.

## Mia40 overexpression reduces the number of cells with Rnq1 aggregation seeds

What is the molecular basis of the different aggregation phenotypes in wild-type and *GAL*-Mia40 cells? To address this, we induced Q97-GFP expression for 4.5 h in both strains, lysed the cells with the non-denaturing detergent NP-40, and subjected the NP-40-soluble (non-aggregated) fraction to proteomics by quantitative mass spectrometry (Fig 5A and B). We excluded the non-soluble fraction because the limited protein solubility prevents reliable protein quantification by proteomics. We observed, as expected, increased levels of Mia40 in the *GAL*-Mia40 strain (about 6 times more than in cells with normal Mia40 levels) and slightly increased levels of a number of mitochondrial proteins, including some Mia40-dependent IMS proteins (Dataset EV1) confirming previous studies (Wrobel *et al*, 2015; Peleh *et al*, 2016). Interestingly, the NP-40-soluble fractions from the *GAL*-Mia40 strain showed altered levels of Rnq1 which is a well-characterized yeast prion protein that serves as crystallization seed in the aggregation process of many proteins, including polyQ proteins (Meriin *et al*, 2002; Duennwald *et al*, 2006; Douglas *et al*, 2009; Wolfe, Ren *et al*, 2013).

The transcription of *RNQ1* was not altered in *GAL*-Mia40 cells as judged from qPCR signals (Fig 5C). We therefore raised Rnq1 antibodies which showed comparable Rnq1 protein levels in both strains. However, a large fraction of Rnq1 formed aggregates in wild-type cells, whereas Rnq1 was soluble in the *GAL*-Mia40 strain (Fig 5D and E).

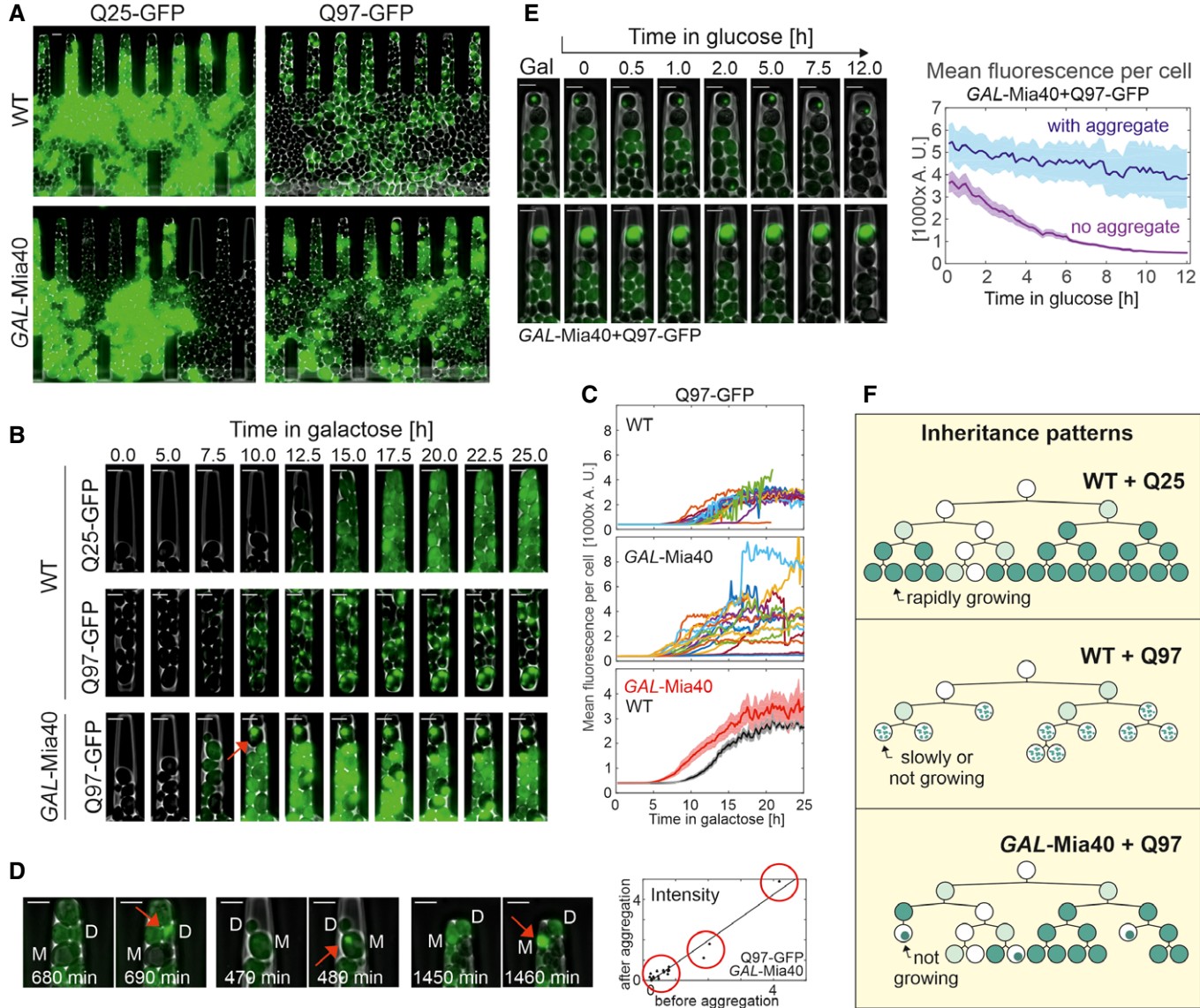

**Figure 4. Q97-GFP shows a sporadic aggregation behavior in *GAL*-Mia40 cells.**

A  Cells were grown in microfluidic chambers (see Movies EV1–EV4). To better visualize the fluorescence signals, different intensity settings were used, so that the intensities here are not comparable. Bars, 4 μm.

B  Still frames after different times of growth indicate a uniform fluorescence in Q25-GFP-expressing cells. Q97-GFP expression in wild type leads to many small and scattered aggregates per cell. In *GAL*-Mia40 cells, Q97-GFP accumulates to much larger intensities but then suddenly collapses into one single aggregate per cell. The arrow depicts aggregate formation. Bars, 4 μm.

C  Quantified signal intensities in cells of the indicated mutants. In top and middle graphs, each curve corresponds to a single cell. Averaged signals (mean + SEM) in bottom graph ($N = 20$ for WT and $N = 17$ cells for *GAL*-Mia40).

D  Still frames directly before and after the Q97-GFP aggregate formation in *GAL*-Mia40 cells. Total cellular fluorescence intensity before aggregation and total intensity in the aggregate (red arrow) were quantified and are shown in the graph on the right hand side, indicating that the entire cellular fluorescent signal collapses into the one single aggregate, irrespective of whether cells contain little, high, or very high amounts of GFP signal (groups indicated by three red circles on the quantification). M, mother and D, daughter. Bars, 4 μm.

E  Cells were shifted from galactose to glucose to stop expression of Q97-GFP. The quantified image shows mean values and SEM from 20 cells with aggregates and 15 cells without aggregate, indicating that soluble Q97-GFP is degraded whereas the aggregated protein is of much higher stability. Bars, 4 μm.

F  Schematic inheritance patterns of polyQ proteins in wild-type and *GAL*-Mia40 cells. Compare also Fig EV2 for more details.

The absence of Rnq1 seeds in *GAL*-Mia40 cells might explain the alteration in Q97-GFP aggregation. Indeed, *Δrnq1* cells showed a comparable or even more pronounced Q97-GFP resistance than the *GAL*-Mia40 mutant (Fig EV3A).

To test whether the increased Mia40 levels were causative for the changed Rnq1 aggregation, we generated newly made versions of the wild type and *GAL*-Mia40 mutants (Fig 5F–I, WT(N) and *GAL*-Mia40(N), respectively). They are genetically identical to the strains

used so far in this study but were freshly made from a glycerol stock of the wild type to ensure an identical behavior of Rnq1 at the start of the experiment. In both of these naive strains, Rnq1 was only detectable in the pellet fraction (Fig 5F) and *GAL*-Mia40(N) showed a considerably lower resistance to Q97-GFP than the previous *GAL*-Mia40 strain (Fig 5G), indicating that the different Rnq1 behavior of these strains strongly contributes to their specific polyQ resistance. However, this *GAL*-Mia40(N) strain could still escape Q97-GFP-mediated growth arrest, whereas the wild-type strain could not

(Fig 5H). When we cultured these newly made strains in galactose-containing medium (without expression of any polyQ protein), we noticed that over time the proportion of soluble Rnq1 increased, a trend that was considerably more pronounced when Mia40 was overexpressed (Fig 5I). Thus, the overexpression of Mia40 increases the resistance against polyQ toxicity. This resistance is largely, but not exclusively, exhibited by Mia40-evoqued effects on the formation of Rnq1 seeds (Fig 5J). This is similar to the situation that was reported before for some cytosolic chaperones, such as Sis1, Sti1, or

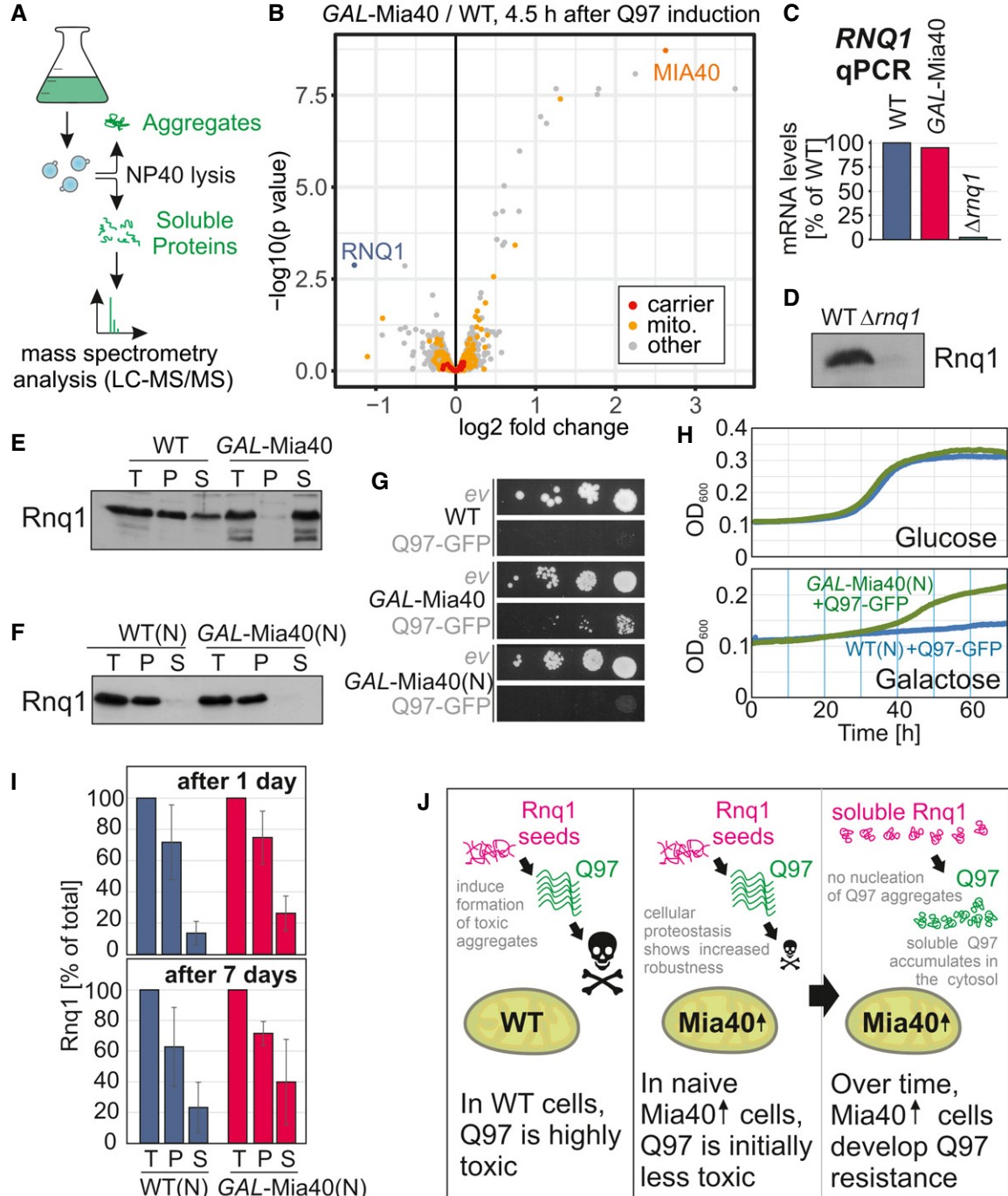

**Figure 5.**

**Figure 5.  Overexpression of Mia40 leads to increased proteostatic robustness of the cytosol.**

A   Workflow of the quantification of soluble proteins by mass spectrometry.

B   Q97-GFP was expressed for 4.5 h in wild-type and *GAL*-Mia40 cells in two biological replicates each. Cells were lysed with NP-40 before soluble proteins were quantified. Mitochondrial proteins (mito.) and members of the mitochondrial carrier family (carrier) are visualized in a volcano plot. *P*-values were derived from a moderated *t*-test. Mitochondrial (mito.) and carrier proteins are indicated by yellow and red dots, respectively. Rnq1 is indicated in blue.

C   Transcript levels for *RNQ1* were measured by qPCR in the strains indicated.

D   Antibodies against Rnq1 were used in Western blots to specifically detect the protein in whole cell extracts of yeast cells.

E, F   Cell extracts of the strains indicated were separated into soluble (S) and aggregated (P) fractions. Rnq1 was detected by Western blotting. Whole cell extracts (total, T) were loaded for control. The wild-type and *GAL*-Mia40(N) strains shown in F were newly generated.

G   Drop dilution experiments of the strains indicated on galactose plates; *ev*, empty vector.

H   Growth curves of the indicated strains at 30°C.

I   Cells of the newly made strains were grown on galactose for 1 or 7 days, respectively. Cells were lysed before the distribution of Rnq1 in the soluble and pellet fractions was quantified. Shown are mean values and standard deviations of three independent replicates.

J   Mia40 overexpression reduces the toxicity of polyQ proteins. In newly made cells, Q97-GFP is still toxic (though less than in wild type) but after several generations, cells escape sensitivity by loss of Rnq1 nucleation seeds.

Source data are available online for this figure.

Hsp104, which likewise reduce polyQ toxicity and likewise reach this at least in part by increasing the solubility of Rnq1 (Meriin *et al*, 2002; Higurashi, Hines *et al*, 2008; Wolfe *et al*, 2013; Klaips *et al*, 2020).

**Cytosolic aggregates interfere with efficient import of mitochondrial proteins**

How can Mia40 influence the propagation of Rnq1 aggregates in yeast cells? Mia40 is a protein of the mitochondrial IMS and there is no indication for any extramitochondrial fraction or aggregation of Mia40, not even upon its overexpression (Fig EV3B and data not shown). Moreover, overexpression of a cytosolic version of Mia40 (i.e., a variant without its mitochondrial targeting sequence) did not suppress the toxicity of Q97-GFP (Fig 6A).

Previous studies showed that cytosolic aggregates can interfere with mitochondrial biogenesis (Li *et al*, 2010; Yano *et al*, 2014; Cenini *et al*, 2016; Lehmer *et al*, 2018; Yablonska *et al*, 2019). We therefore tested whether the expression of Q97-GFP impaired the Mia40-dependent protein import. Since only very low levels of Q97-GFP were coisolated with mitochondria, we avoided *in vitro* import experiments. The import rate of Mia40 substrates such as Cox19 can be analyzed *in vivo* by radiolabeling newly synthesized proteins during a short pulse and subsequent detection of disulfide bond formation during a chase incubation (Fischer, Horn *et al*, 2013). Even high levels of Q97-GFP did not interfere with Mia40-mediated Cox19 import (Fig EV3C), so that a direct effect of polyQ aggregates on Mia40 function appeared unlikely. We also did not observe accumulated precursor proteins of the presequence-containing proteins Mdj1 and Rip1 by Western blotting (Fig 6B), which are, for example, found upon competitive inhibition of mitochondrial protein import by clogger proteins (Boos *et al*, 2019) or in mutants of the mitochondrial processing peptidase (Poveda-Huertes *et al*, 2020).

We recently developed a sensitive *in vivo* assay to detect the cytosolic accumulation of mitochondrial precursor proteins (Hansen, Aviram *et al*, 2018) (Fig EV3D). Expression of a fusion protein consisting of the nuclear encoded mitochondrial protein Oxa1 and the cytosolic uracil biosynthesis enzyme Ura3 in a *Δura3* strain leads to uracil-dependent growth as long as mitochondrial protein import is fully functional, but to uracil independence when import is inefficient (Fig EV3D). When Oxa1-Ura3 was co-expressed with Q97-GFP (from a constitutive medium-expression *GPD* promoter), cells became uracil-independent, indicative for the cytosolic accumulation of the Oxa1-Ura3 fusion protein. Similarly, the expression of other aggregation-prone proteins, such as a disease variant of TDP-43, FUS, or α-synuclein, also rendered cells uracil-independent, whereas non-aggregating proteins, such as Q25-GFP, TDP-43, or GFP, did not (Fig 6C). Obviously, cytosolic aggregates of different kind lead to the cytosolic accumulation of the Oxa1-Ura3 reporter, either by interference with its mitochondrial import or with its proteolytic degradation in the cytosol.

**Figure 6.  Cytosolic polyQ aggregates interfere with efficient protein import into mitochondria.**

A   Wild-type and *GAL*-Mia40 cells, and cells expressing a cytosolic version of Mia40 under *GAL1* control were dropped on the indicated plates; *ev*, empty vector. Note that the mitochondrial version of Mia40 suppressed polyQ toxicity, whereas the cytosolic version did not.

B   In wild-type or *GAL*-Mia40 cells, Q25-GFP or Q97-GFP was expressed for 4.5 h. For control, wild-type cells were used which expressed also for 4.5 h the "clogger" protein cytochrome b$_2$-DHFR, a strong competitive inhibitor of protein import. Samples were analyzed by Western blotting. The presence of precursor proteins (labeled as *P*) in addition to the mature forms (labeled as m) is indicative of a strong import defect.

C   *Δura3* cells expressing the Oxa1-Ura3 reporter for the cytosolic accumulation of mitochondrial precursor proteins (Hansen *et al*, 2018) were transformed with plasmids expressing the proteins TDP-43, the TDP-43$^{Q331K}$ mutant, Q25-GFP, Q97-GFP, α-synuclein, FUS, or GFP from a constitutive glyceraldehyde-3-phosphate dehydrogenase (GPD) promoter. Note that the expression of all aggregation-prone proteins (indicated by red arrows) allows cells to grow on uracil-deficient plates indicating the cytosolic accumulation of the Oxa1-Ura3 fusion protein.

D, E   Wild-type cells were transformed with plasmids for the expression of the indicated proteins. Q97-GFPnF is a non-fluorescent version of Q97-GFP. Close proximity of the two split-GFP parts results in fluorescence which was quantified in a spectrometer and visualized by microscopy. Shown are mean values and standard deviations of three replicates. Bar, 5 μm.

Source data are available online for this figure.

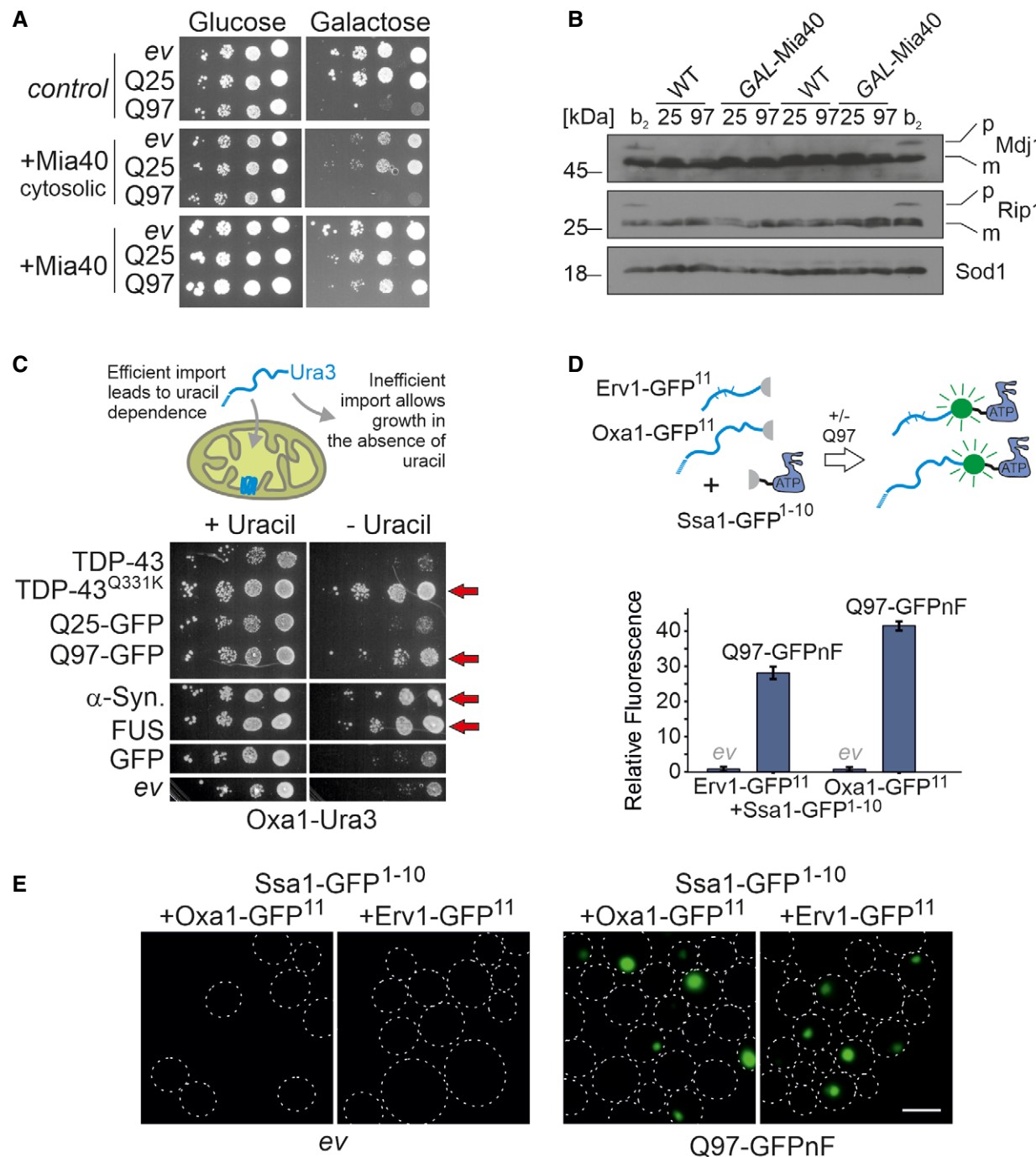

**Figure 6.**

To more directly visualize the accumulation of cytosolic precursor proteins, we employed a fluorescence-based screen using a self-complementing split version of superfolder GFP (Pedelacq, Cabantous *et al*, 2006; Smoyer, Katta *et al*, 2016; Laborenz, Bykov *et al*, 2021). We fused one part of GFP (the C-terminal beta sheet consisting of 17 residues) to the C-terminus of Oxa1 (Oxa1-GFP[11]) or the Mia40 substrate Erv1 (Erv1-GFP[11]) and the other part of GFP to the cytosolic Hsp70 protein Ssa1 (Ssa1-GFP[1-10]). These proteins were constitutively expressed from plasmids. Moreover, we

generated a non-fluorescent Q97-GFP version (Fig EV3E, Q97-GFPnF) that did not interfere with the use of the split-GFP reporter. Upon expression of Q97-GFPnF for 4.5 h, both split-GFP pairs showed a strong fluorescence (Fig 6D). These signals were emitted from individual puncta in the cells, indicating that Ssa1 encounters Erv1 and Oxa1 in defined locations in the cytosol (Fig 6E). In summary, our results indicate that the expression of Q97-GFP leads to the cytosolic accumulation of some mitochondrial precursor proteins which co-localize with the cytosolic Hsp70 protein Ssa1.

### The biogenesis of mitochondrial membrane proteins challenges cytosolic proteostasis

For a more general and unbiased overview, we quantitatively analyzed the proteomes of wild-type cells 4.5 h after induction of either Q25-GFP or Q97-GFP (Fig 7A). This *in vivo* method indicated a weak, but consistent reduction of carrier proteins, i.e., hydrophobic inner membrane proteins. Taken the relatively short time of polyQ expression (correlating to one or two cell divisions) into account, such a drop in total protein levels could point to a considerable reduction of the biogenesis of these proteins.

Does also the overexpression of other components of the mitochondrial import machinery suppress the polyQ toxicity? To test this, we cloned the sequences for 10 other mitochondrial proteins as well as that of Hsp104, a well-characterized suppressor of polyQ toxicity (Fig EV4A and B), into the *GAL* plasmid and co-expressed these proteins with Q97-GFP. Interestingly, the overexpression of several mitochondrial proteins (Erv1, Tim9, Sam50, and Sam37)

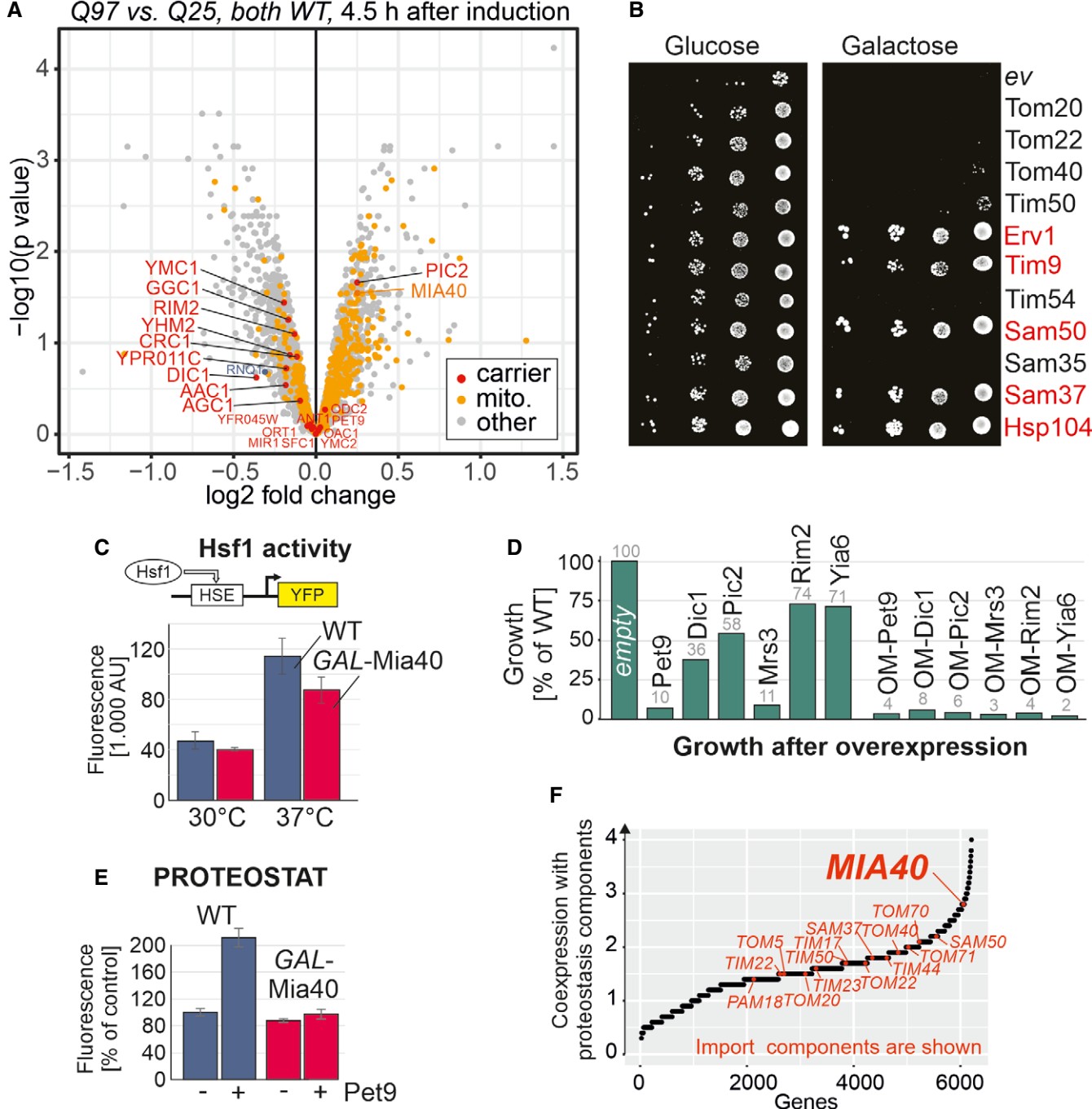

**Figure 7.**

**Figure 7.  Mitochondrial membrane proteins can pose a threat to cytosolic proteostasis.**

A   Q25-GFP and Q97-GFP were expressed in wild-type cells for 4.5 h. Cell extracts were subjected to mass spectrometry-based proteomics. The levels of Mia40 and other mitochondrial proteins (mito., yellow) and carrier proteins (orange) are visualized in a volcano plot (*P*-values were derived using a moderated *t*-test). The position of Rnq1 is indicated in blue.

B   The open reading frames of the indicated proteins were cloned downstream of a *GAL* promoter in plasmids and transformed into cells harboring the Q97-GFP expression plasmid. Cells harboring an empty vector (*ev*) were used for control. Cells were shifted to galactose medium overnight before they were dropped onto glucose or galactose plates. If cells are directly dropped from lactate medium onto galactose plates, only individual cells (indicated by red protein names) escaped the polyQ toxicity (see Fig EV4C).

C   Wild-type and *GAL*-Mia40 cells were transformed with plasmids that express yellow fluorescent protein (YFP) under the control of a minimal promoter containing the Hsf1-driven heat shock element (HSE). Cells were grown at 30 or 37°C before fluorescence was measured. Mean values and standard deviations of three replicates are shown.

D   Cells expressing the indicated carrier proteins from *GAL* promoters were dropped onto galactose plates. Cell growth was quantified by densitometry from one representative plate. OM refers to carrier proteins that were expressed as fusions with an N-terminal outer membrane anchor. See Fig EV4 D and E for scans of the respective plates.

E   The formation of protein aggregates was detected with a PROTEOSTAT protein aggregation assay. The expression of the most abundant mitochondrial carrier protein, Pet9, resulted in a strongly increased fluorescence in wild-type cells but not in *GAL*-Mia40 cells. Mean values and standard deviations of three replicates are shown.

F   Using the SPELL data collection of transcription data sets, we calculated the co-expression of yeast genes with genes relevant for proteostasis (chaperones, proteasome, and others; see Materials and Methods for details). Genes for mitochondrial import components are highlighted.

Source data are available online for this figure.

suppressed Q97-GFP toxicity, whereas we did not observe suppression by overexpression of Tom20, Tom22, Tom40, Tim50, and Tim54 (Fig 7B). The uniform and robust growth was only observed when precultures were already exposed to galactose. However, when cells were directly dropped from lactate medium onto galactose plates, the growth was heterogeneous (Fig EV4C), indicating that the cells need first to adapt to become polyQ-resistant. This behavior was also observed with Hsp104. It appears likely that this heterogeneity is again caused by Rnq1, similar to what we had observed for the Mia40-mediated suppression.

Why do elevated levels of Mia40 and other import factors provide resistance against proteotoxic stress? Previous studies already showed that Mia40 is rate-limiting for the import of its client proteins into mitochondria and that increased Mia40 levels result in increased steady state levels of its substrates (Peleh *et al*, 2016). Moreover, it was shown that the Mia40-dependent import process directly competes with proteasomal degradation (Kowalski, Bragoszewski *et al*, 2018; Mohanraj, Wasilewski *et al*, 2019). It therefore seems conceivable that increased import rates into mitochondria reduce the burden on cytosolic proteostasis. In order to test this, we made use of a reporter system that measures the heat shock response by Hsf1-driven expression of a fluorescent reporter. Growth at 37°C induced the heat shock response to a larger extent in wild type than in *GAL*-Mia40 cells suggesting that this strain indeed is less reactive to heat stress (Fig 7C).

Presumably owing to their hydrophobic nature, carrier proteins were identified as being a severe burden for cytosolic proteostasis when their mitochondrial import is impaired (Wang & Chen, 2015; Piard, Umanah *et al*, 2018; Hoshino, Wang *et al*, 2019; Liu, Wang *et al*, 2019; Backes, Bykov *et al*, 2021). Indeed, we observed that overexpression of carrier proteins is highly toxic to cells, in particular when they are expressed as fusion proteins with an outer membrane anchor (Figs 7D and EV4D–F).

We employed an established assay of PROTEOSTAT staining (Pihlasalo, Kirjavainen *et al*, 2011) to monitor the formation of misfolded proteins in yeast cells. Overexpression of the ATP/ADP carrier Pet9 indeed strongly increased the PROTEOSTAT signal (indicating protein misfolding), but did hardly change the signal in

*GAL*-Mia40 cells (Fig 7E). From this, we conclude that high levels of Mia40 can be protective against the toxic effects of mitochondrial precursor proteins. Interestingly, previous studies have shown that cells induce Mia40 when cytosolic precursors accumulate (Boos *et al*, 2019) and the expression of Mia40 is co-regulated with that of components of the chaperone and proteasome system, in contrast to other mitochondrial import components (Fig 7F). However, Mia40 overexpression does not generally provide increased stress resistance and can either be protective or sensitizing depending on the specific nature of the applied stress conditions (Fig EV5A).

### Mia40 overexpression also influences Q97-GFP aggregation in a mammalian cell culture model

Is the strong suppression of polyQ toxicity by Mia40 a peculiarity of the yeast model system? To address this question, we first verified that the expression of human MIA40 in human embryonic kidney (HEK293) cells changed the mitochondrial proteome. Western blotting (Fig 8A) as well as quantitative mass spectrometry (Habich *et al*, 2019) confirmed that higher levels of MIA40 also increase the steady state levels of some mitochondrial proteins, even though not to the same extent as in yeast (Wrobel *et al*, 2015; Peleh *et al*, 2016).

Next, we employed an established cell culture model using human neuroblastoma SH-SY5Y cells that expressed Q97-GFP. This system has been extensively used before as a model for Huntington's disease (van Well, Bader *et al*, 2019). In many of these cells, Q97-GFP formed aggregates that often appeared in proximity to mitochondria (Figs 8B and EV5B). Co-expression of MIA40 in these cells significantly decreased the number of cells with Q97-GFP aggregates (Fig 8B and C). This suppression of polyQ aggregation was not seen when a variant of MIA40 (ΔN-MIA40) was used which lacked the N-terminal 40 amino acid residues that are important for efficient mitochondrial targeting (Hangen, Feraud *et al*, 2015). We also noticed that the co-expression of MIA40 and Q97-GFP in SH-SY5Y cells induced changes in the mitochondrial morphology as many mitochondria formed large circular structures (Figs 8D and EV5C). These structures presumably represent dilated mitochondria

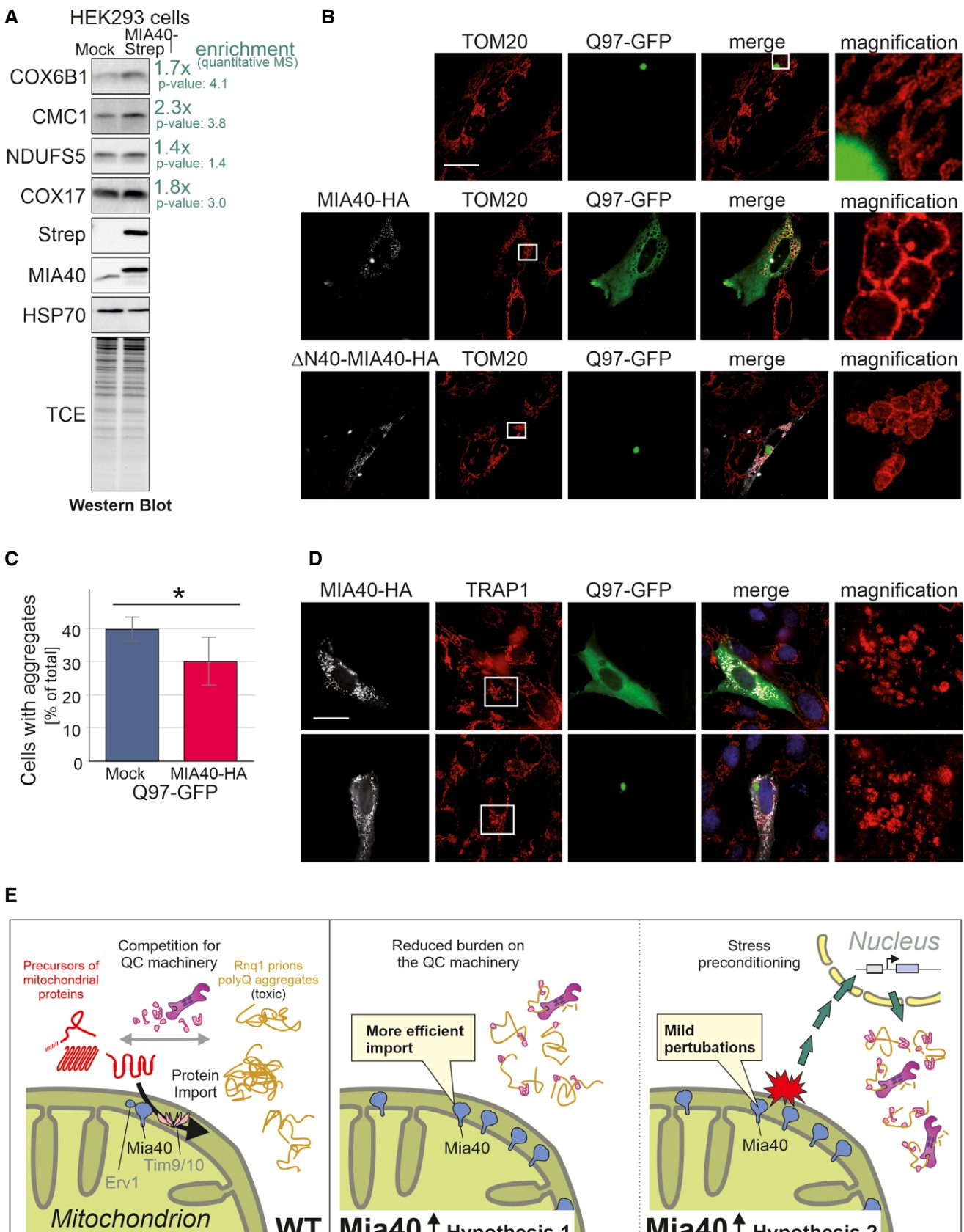

**Figure 8.**

◀

**Figure 8. Overexpression of MIA40 reduces the number of human SH-SY5Y cells with polyQ aggregates.**

A   Cell extracts of HEK293 cells expressing a C-terminally Strep-tagged version of MIA40 (MIA40-Strep) or a mock control were analyzed by Western blotting with antibodies against several IMS proteins as well as HSP70 which served as loading control. The enrichment factors shown in green are based on quantitative mass spectrometry data (Habich *et al*, 2019). *P*-values were derived from Student's *t*-test. TCE, trichloroethanol protein stain.

B   SH-SY5Y cells co-expressing Q97-GFP with MIA40 (middle row), ΔN-MIA40 (lower row), or a mock-transfected control (upper row) were analyzed by immunocytochemistry and super-resolution microscopy. The magnification shows an enlarged section of the TOM20 staining. TOM20 and TRAP1 serve as markers for the mitochondrial outer membrane and the matrix, respectively. Bars, 20 μm. Magnifications: 40-fold of a 63x objective field.

C   The percentage of cells with Q97-GFP aggregates was quantified in control (mock transfected) and MIA40-expressing cells. Data are displayed as mean ± standard deviations and were analyzed by two-tailed Student's *t*-test, *n* = 6. *$P \leq 0.05$.

D   SH-SY5Y cells co-expressing Q97-GFP with MIA40 (upper row) or ΔN-MIA40 (lower row) were analyzed by immunocytochemistry and super-resolution microscopy. The magnification shows an enlarged section of the TRAP1 staining used as a marker for the mitochondrial matrix, respectively. Bars, 20 μm. Magnifications: 40-fold of a 63× objective field.

E   Model for competition of mitochondrial precursor proteins and aggregation-prone proteins for factors of the quality control (QC) machinery, such as chaperones and components of the proteasome system. Our observations indicate that mitochondrial membrane proteins pose a particular threat to cytosolic proteostasis. Overexpression of the indicated components of the import machinery suppresses the toxicity of polyQ proteins for which the prion protein Rnq1 plays a critical role.

Source data are available online for this figure.

as their lumen contained the soluble matrix protein TRAP1. Such alterations were not seen if either MIA40 or Q97-GFP were individually expressed or when MIA40 was co-expressed with Q25-GFP (Fig EV5D). These morphological changes were reminiscent of the "curly" mitochondria that were induced in yeast cells by co-expression of Mia40 and Q97-GFP (Fig EV1).

Thus, increased levels of MIA40 can also be protective against the formation of polyQ aggregates in mammalian cells. Obviously, the stability of cytosolic proteostasis and the performance of the mitochondrial protein import machinery are generally connected in eukaryotic cells.

# Discussion

Mitochondria can make up a large fraction of the cellular mass (more than 30% in some cell types) and dynamically adapt their volume to metabolic conditions. Owing to their post-translational mode of import, newly synthesized mitochondrial precursors initially explore the cytosol where they are maintained in a soluble and import-competent state by different chaperones that reside in the cytosol or are associated with the membranes of mitochondria and the ER (Deshaies, Koch *et al*, 1988; Becker, Walter *et al*, 1996; Terada, Kanazawa *et al*, 1997; Fünfschilling & Rospert, 1999; Papic, Elbaz-Alon *et al*, 2013; Hoseini, Pandey *et al*, 2016; Hansen *et al*, 2018; Jores, Lawatscheck *et al*, 2018; Opalinski, Song *et al*, 2018). Thereby, mitochondrial protein biogenesis directly depends on the cytosolic chaperone capacity. Presumably as a consequence of their strong tendency to sequester chaperones, the cytosolic accumulation of precursor proteins induces a sudden growth arrest and triggers the increased expression of components of the chaperone and proteasome system (Wang & Chen, 2015; Wrobel *et al*, 2015; Weidberg & Amon, 2018; Boos *et al*, 2019; Mårtensson, Priesnitz *et al*, 2019; Boos, Labbadia *et al*, 2020; Shakya *et al*, 2021). Obviously, the post-translational import mode of mitochondrial proteins poses a threat for cellular proteostasis which is met by an adaptive network of cytosolic factors that can deal with unfolded precursors.

Our observations show that the synthesis, targeting, and import of mitochondrial proteins are of direct relevance for the aggregation behavior of cytosolic polyQ proteins. We found that the *mia40-3* mutant is hypersensitive to polyQ proteins even at permissive

growth conditions at which no mitochondrial defects are apparent in this strain (Chacinska *et al*, 2004). Temperature-sensitive Mia40 mutants were instrumental to identify an increased proteasomal capacity as reaction to a reduced mitochondrial import efficiency, a program that is referred to as unfolded protein response activated by mistargeting of proteins (UPRam)(Wrobel *et al*, 2015). The hypersensitivity of this mutant suggests that the simultaneous presence of accumulated precursors and of polyQ proteins leads to a collapse of proteostasis in these mutants. Apparently, both groups of problematic proteins put a strain on the same components of the chaperone and protein quality control network (Fig 8E). In this context, mitochondrial membrane proteins, in particular the very abundant members of the carrier family, are highly relevant. The pronounced toxicity of this group of proteins was noticed also in previous studies (Vartiainen, Chen *et al*, 2014; Wang & Chen, 2015; Hoshino *et al*, 2019; Liu *et al*, 2019; Backes *et al*, 2021). Many mitochondrial membrane proteins, including carriers and the beta barrel proteins of the outer membrane, are synthesized without presequences, and the kinetics of their import reaction is not known. However, it is conceivable that the accumulation of these proteins in the cytosol and, even more, on the mitochondrial surface is problematic as these proteins might sequester cytosolic chaperones or serve as nucleation sites for the formation of aggregates (Jores *et al*, 2018; Opalinski *et al*, 2018). The relevance of mitochondrial surfaces in the context of aggregate formation was described in yeast and human cells (Aguilaniu *et al*, 2003; Liu *et al*, 2004; Ferri *et al*, 2006; Gruber *et al*, 2018; Suhm *et al*, 2018; Yablonska *et al*, 2019).

It was still surprising to see that the overexpression of Mia40 is protective against cytosolic aggregates. How can this be explained? Two mutually not exclusive mechanisms seem plausible:

1   It was shown in previous studies that Mia40 is rate-limiting for mitochondrial protein import. Increased levels of Mia40 lead to increased protein levels in yeast or human mitochondria (Wrobel *et al*, 2015; Peleh *et al*, 2016; Habich *et al*, 2019). Thus, at normal Mia40 levels, a fraction of newly synthesized mitochondrial proteins is apparently never imported but retained in the cytosol and finally degraded. Recent studies indeed directly demonstrated the competition of the Mia40-mediated import and proteasomal degradation (Kowalski *et al*, 2018; Mohanraj

*et al*, 2019; Krämer, Groh *et al*, 2020). Thus, increased Mia40 levels presumably speed up the import of certain proteins and thus might reduce their burden on cytosolic proteostasis (Fig 8E, middle panel).

2   The Mia40 pathway is only one out of several import routes into mitochondria. It is not unlikely that the overexpression of Mia40 might have negative effects on clients of these other import routes. Indeed, we observed that very high levels of Mia40, even higher than those used in this study, are toxic. Thus, Mia40 overexpression might induce mild perturbations that result in cellular adaptations which give rise to the observed polyQ resistance. Preconditioning to stress conditions is well known from the heat stress response where preexposure to elevated temperature provides resistance to heat. At least in worms, sublethal mitochondrial stress induces a latent heat shock response which is protective against proteostasis collapse (Labbadia *et al*, 2017; Wu, Senchuk *et al*, 2018; Zhang, Wu *et al*, 2018). Such mild Mia40-evoqued stress conditions might also explain the increased elimination of Rnq1 aggregates by counterselection (Fig 8E, right panel).

The observation that also overexpression of other mitochondrial import factors suppresses polyQ toxicity is compatible with both hypotheses. Apparently, increased levels of several factors of the carrier and the beta barrel protein import pathway induce alterations in the cytosolic proteostasis network which provide resistance against specific proteotoxic stress conditions.

Is the modulation of Mia40 levels relevant under physiological conditions? In contrast to other components of the mitochondrial import machinery, the expression of Mia40 is upregulated upon stress conditions induced by mitochondrial defects (Boos *et al*, 2019; Zöller, Laborenz *et al*, 2020) but also by problems outside mitochondria (Metzger & Michaelis, 2009). Transcription of *MIA40* is under control of Rpn4, a transcription factor that adjusts proteasome levels (Xie & Varshavsky, 2001). Most substrates of Mia40 are small hydrophilic proteins of the IMS that are unlikely to be harmful for the cytosol. However, major clients of the Mia40 pathway include the small Tim proteins, which are essential components for the import of inner membrane carriers (Koehler, Jarosch *et al*, 1998; Luciano, Vial *et al*, 2001). Owing to their abundance and hydrophobicity, misfolded or non-imported carriers are known to pose a major threat to cytosolic proteostasis. In consistence with our results, a previous genome-wide overexpression screen for components relevant in the context of polyQ toxicity identified components of the carrier import pathway as suppressors of polyQ-induced growth defects, including Erv1, Tim9, Tim10, Tim12, and Tim22 (Mason *et al*, 2013). It will be very exciting to study the early targeting steps of carrier proteins and their impact on cytosolic proteostasis in more detail.

Our study documents that modulation of the capacity of the mitochondrial import machinery is of direct relevance for cytosolic proteostasis. Upregulation of mitochondrial import components, in particular of Mia40, might provide a strategy to stabilize aggregation-prone proteins in human cells. The well-known decline of cytosolic proteostasis in aging cells, and in particular in neurons of patients with Huntington's disease, along with slower import rates into mitochondria would increase the abundance of cytosolic precursors and the occupancy of precursor-binding chaperones. The

upregulation of Mia40 levels or boosting the functionality of the mitochondrial disulfide relay may provide a strategy to better stabilize cytosolic proteostasis in such cells. It will be exciting to explore this in the future.

## Materials and Methods

### Yeast strains and plasmids

All yeast strains used in this study are based on the wild-type strains BY4742 (MATα *his3 leu2 lys2 ura3*) and YPH499 (MATa *ura3 lys2 ade2 trp1 his3 leu2*). They are listed in Table EV1. The *mia40-3* mutant (Chacinska *et al*, 2004) and the *GAL*-Mia40 (Terziyska, Lutz *et al*, 2005) strain were described previously. For the generation of *GAL*-Mia40(N), the sequence of the *HIS3* gene and the *GAL10* promoter was amplified from the plasmid pTL26 and inserted upstream of *MIA40*. To delete *RNQ1* in YPH499 and *GAL*-Mia40, the kanMX4 cassette for G418 resistance was amplified from pFA6a-kanMX4 and genomically integrated by homologous recombination.

For the Hsf1 activity reporter, the pNH605-4xHSEpr-YFP reporter plasmid was used as described before (Zheng, Krakowiak *et al*, 2016). To generate a Pet9 overexpression plasmid, the coding region of *PET9* was amplified from genomic DNA and ligated into a pYX233 empty vector plasmid between the *GAL1* promoter and an HA tag by using the restriction sites *Eco*RI and *Xma*I. To generate pET19b-Rnq1, the *RNQ1* ORF was amplified from isolated YPH499 gDNA using primers for the introduction of *Nde*I and *Bam*HI restriction sites. After restriction digest of the PCR product and the pET19b vector, ligation was performed to generate the pET19b-Rnq1 plasmid for the bacterial expression of N-terminally His-tagged Rnq1.

To generate Entry Clones and Expression Clones, the Gateway technology was applied according to the instructions of the manufacturer (Invitrogen). BP reactions and LR reactions were used basically as described in the Gateway technology user guide. To generate Entry Clones, the ORF of interest was amplified by PCR using YPH499 genomic DNA as template and primers for the introduction of *att*B-sites. The resulting *att*B-PCR product was mixed with the pDONR221 Donor Vector in a BP reaction. To generate Expression Clones, the Entry Clone carrying the ORF of interest was mixed with the Destination Vector containing the desired properties for expression in yeast in an LR reaction.

To generate a non-fluorescent version of Q97-GFP, the glycine at position 67 of the eGFP amino acid sequence was changed to alanine in the respective Entry Clone by site-directed mutagenesis. To this end, primers to introduce a G to C mutation were designed to be 28 nucleotides long and to carry the desired mutation in the middle of the primer. The Entry Clone was then subjected to error-prone PCR using the indicated primers and the PfuUltra HF DNA polymerase. The template DNA was removed after the reaction by digestion with *Dpn*I. The same cloning strategy was used to generate a non-fluorescent version of Q25-GFP. After sequencing of the Entry Clones, an LR reaction was performed to produce the pAG424GAL-Q25-EGFPnF and pAG424GAL-Q97-EGFPnF plasmids.

Strains were grown in yeast complete medium (1% yeast extract, 2% peptone), containing 2% of the carbon sources galactose or glucose as indicated. The temperature-sensitive strain (*mia40-3*) was grown at 25°C before switching it to 30°C. Strains containing

plasmids were grown at 30°C in minimal synthetic medium containing 0.67% yeast nitrogen base and 2% lactate as carbon source. To induce the expression from the *GAL1* promoter, cultures were supplemented with 2% galactose.

### Human cell lines and plasmids

For transfection, 0.15–0.3 μg of DNA per 24-well (150,000 cells plated) was mixed with 50 μl Opti-MEM and 4 μl polybrene in tube A. Meanwhile, 1 μl lipofectamine was added to tube B which also contained 50 μl Opti-MEM. Both tubes were incubated for 15 min. The contents of both tubes were further incubated for 15 min. Cells were washed with PBS (−/−), and 500 μl of Opti-MEM was added to each well. Following the 15-min incubation, 100 μl of the mix was added dropwise to each well.

### Culture conditions of human cells

SH-SY5Y cells were kept in a humidified incubator at 37°C, 5% CO2. Passage of the cells was performed as follows: Cells were washed with PBS (−/−), and then, 3 ml of trypsin was added to a 175-cm$^2$ flask and incubated for 2 min in the incubator. Trypsinized cells were resuspended in 7 ml medium (DMEM F12 + 15% fetal calf serum + 1% penicillin/streptomycin + 1% non-essential amino acids (NEAA); obtained from Gibco, Invitrogen) and centrifuged. The cell pellet was again resuspended in 10 ml medium and divided in flasks in a proportion of 1/10.

### Viability assay

Yeast cells were grown to mid-log phase in lactate medium before the induction of Q97-GFP was induced by the addition of 2% galactose. Samples were taken after different times and diluted to OD (600 nm) of 0.0002 in 300 μl of sterile water. 80 μl of the cell suspension was plated in technical triplicates on glucose-containing plates to repress the expression of Q97-GFP. The plates were incubated for 5 days to assess the number of colonies which survived the expression of Q97-GFP.

### Subcellular fractionation analysis of Rnq1

To determine the aggregation state of Rnq1, a previously established protocol was followed (Sideri, Koloteva-Levine et al, 2011). Cells were grown until log phase. 5.0 OD (600 nm) were harvested and washed once with distilled water. The cell pellet was resuspended in 200 μl of lysis buffer (74 mM Tris–HCl pH 8.0, 150 mM KCl, 50 mM EDTA, 1 mM DTT, 0.2% SDS, 1% Triton X-100, 1 mM PMSF, 1x cOmplete Mini Protease Inhibitor mixture (Roche)) and transferred to a new tube containing 200 μl of glass beads for mechanical lysis using a FastPrep bead beater. To remove cell debris, the lysates were centrifuged for 3 min at 4°C and 800 g. 50 μl of the supernatant was added to 50 μl of reducing sample buffer as the total. Another 50 μl was transferred to a new tube and centrifuged for 1 h at 85,000 g, 4°C, to generate supernatant and pellet fractions. The supernatant was transferred to a new tube, and the pellet was resuspended in 50 μl of lysis buffer. 50 μl of reducing sample buffer was added to both the supernatant and the pellet fraction. All samples were incubated at 96°C for 4 min and

subsequently subjected to SDS–PAGE and Western Blot using an anti-Rnq1 antibody for the detection of Rnq1 in either the pellet or the supernatant fraction.

### Sample preparation, mass spectrometry, and proteomics data analysis

To compare the soluble proteomes of wild-type and *GAL*-Mia40 yeast strains which both have been expressing Q97-GFP for 4.5 h, as well as the changes in the proteomes of WT strains expressing either Q97-GFP or Q25-GFP, the expression of the polyQ constructs was induced for 0 h and 4.5 h. 50 ml of mid-log phase cell suspension was harvested by centrifugation at RT, 4,000 g for 5 min and washed twice with PBS before freezing the pellet in liquid nitrogen and storing it at −80°C. The cells were resuspended in PBS with a volume in ml equal to $0.4 \times OD_{600}$ (e.g., if $OD_{600}$ = 0.5, add 200 μl; calculation: $0.5 \times 0.4 = 0.2$ ml). 40 μl of the cell suspension was transferred to a new tube to which 60 μl of cold 1.7× lysis buffer (0.8% NP-40 in PBS, 1 mM $MgCl_2$, 0.25 U/μl benzonase, cOmplete Protease Inhibitor Cocktail, Phosphatase Inhibitor, PBS) was added. Cell lysates were prepared using a FastPrep-24 5G homogenizer (MP Biomedicals, Heidelberg, Germany) with 3 cycles of 30 s, speed 8.0 m/s, 120 s breaks, glass beads. To remove cell debris, the lysates were centrifuged at 4°C for 10 min at 16,100 g. 40 μl of the supernatant was transferred onto a pre-wet 0.45-μm filter plate on a 96-well plate and centrifuged for 5 min at 500 × g, 4°C to collect the lysate free of Q97-GFP aggregates. The sample plate was sealed with a plate seal plastic foil, dipped in liquid nitrogen, and frozen at −80°C until use.

For LC-MS/MS analysis of lysates, protein concentration of lysates was determined by BCA protein determination. 10 μg of each lysate was subjected to an in-solution tryptic digest using a modified version of the Single-Pot Solid-Phase-enhanced Sample Preparation (SP3) protocol (Hughes, Foehr et al, 2014; Moggridge, Sorensen et al, 2018). Lysates were added to Sera-Mag Beads (Thermo Scientific) in 10 μl 15% formic acid and 30 μl of ethanol. Binding of proteins was achieved by shaking for 15 min at room temperature. SDS was removed by 4 subsequent washes with 200 μl of 70% ethanol. Proteins were digested overnight at room temperature with 0.4 μg of sequencing grade modified trypsin (Promega) in 40 μl Hepes/NaOH, pH 8.4 in the presence of 1.25 mM TCEP and 5 mM chloroacetamide (Sigma-Aldrich). Beads were separated and washed with 10 μl of an aqueous solution of 2% DMSO, and the combined eluates were dried down. Peptides were reconstituted in 10 μl of $H_2O$ and reacted for 1 h at room temperature with 80 μg of TMT10plex (Thermo Scientific) (Werner, Sweetman et al, 2014) label reagent dissolved in 4 μl of acetonitrile. Excess TMT reagent was quenched by the addition of 4 μl of an aqueous 5% hydroxylamine solution. Peptides were reconstituted in 0.1% formic acid, mixed to achieve a 1:1 ratio across all TMT channels, and purified by a reverse phase clean-up step (OASIS HLB 96-well μElution Plate, Waters). Peptides were subjected to an off-line fractionation under high pH conditions (Hughes et al, 2014). The resulting 12 fractions were then analyzed by LC-MS/MS on an Orbitrap Fusion Lumos mass spectrometer (Thermo Scientific) as previously described (Sridharan, Kurzawa et al, 2019). To this end, peptides were separated using an Ultimate 3000 nano RSLC system (Dionex) equipped with a trapping cartridge (Precolumn C18 PepMap100, 5 mm, 300 μm i.d., 5 μm, 100 Å) and an analytical column (Acclaim

PepMap 100. 75 × 50 cm C18, 3 mm, 100 Å) connected to a nanospray-Flex ion source. The peptides were loaded onto the trap column at 30 µl per min using solvent A (0.1% formic acid) and eluted using a gradient from 2 to 40% Solvent B (0.1% formic acid in acetonitrile) over 2 h at 0.3 µl per min (all solvents were of LC-MS grade). The Orbitrap Fusion Lumos was operated in positive ion mode with a spray voltage of 2.4 kV and capillary temperature of 275°C. Full scan MS spectra with a mass range of 375–1,500 $m/z$ were acquired in profile mode using a resolution of 120,000 (maximum fill time of 50 ms or a maximum of 4e5 ions (AGC) and a RF lens setting of 30%. Fragmentation was triggered for 3 s cycle time for peptide like features with charge states of 2–7 on the MS scan (data-dependent acquisition). Precursors were isolated using the quadrupole with a window of 0.7 $m/z$ and fragmented with a normalized collision energy of 38. Fragment mass spectra were acquired in profile mode and a resolution of 30,000 in profile mode. Maximum fill time was set to 64 ms or an AGC target of 1e5 ions). The dynamic exclusion was set to 45 s.

Acquired data were analyzed using IsobarQuant (Franken, Mathieson *et al*, 2015) and Mascot V2.4 (Matrix Science) using a reverse UniProt FASTA *Saccharomyces cerevisiae* database (UP000002311) including common contaminants and the two protein sequences of Q25-GFP and Q97-GFP.

The following modifications were taken into account: Carbamidomethyl (C, fixed), TMT10plex (K, fixed), Acetyl (N-term, variable), Oxidation (M, variable), and TMT10plex (N-term, variable). The mass error tolerance for full scan MS spectra was set to 10 ppm and for MS/MS spectra to 0.02 Da. A maximum of 2 missed cleavages were allowed. A minimum of 2 unique peptides with a peptide length of at least seven amino acids and a false discovery rate below 0.01 were required on the peptide and protein level (Savitski, Wilhelm *et al*, 2015).

The raw output files of IsobarQuant (protein.txt—files) were processed using the R programming language (ISBN 3-900051-07-0). Only proteins that were quantified with at least two unique peptides were considered for the analysis. Moreover, only proteins which were identified in both mass spec runs were kept. 2915 proteins passed these quality control filters. Raw TMT reporter ion signals (signal_sum columns) were first cleaned for batch effects using the "removeBatchEffect" function of the limma package (Ritchie, Phipson *et al*, 2015) and further normalized using vsn (variance stabilization normalization) (Huber *et al*, 2002). Missing values were imputed with knn method using the Msnbase package (Gatto & Lilley, 2012). Proteins were tested for differential expression using the limma package. T-value outputs of the limma package were used as an input for the fdrtool function of the fdrtool package (Strimmer, 2008) in order to estimate *P*-values and false discovery rates (qvalues were used). Proteins were classified as "hit" with an fdr smaller 5% and a fold-change of at least 50% and classified as "candidate" with an fdr smaller 20% and a fold-change of at least 40% using either the fdr estimated by limma directly or by the fdrtool package, depending on which method led to a faster approach to zero of the fdr with increasing effect size (t value).

## Superfolder split-GFP assay

Cells contain the plasmids pYX142-Oxa1-GFP11 or pYX142-Erv1-GFP11 in combination with pYX122-Ssa1-GFP1-10 or pYX122-Mia40-GFP1-10 and also either pAG424GAL-Q97-EGFPnF or the control plasmid pAG424GAL-ccdB. Cells were grown in selective medium containing 2% lactate to mid-log phase. Expression of Q97-GFPnF was induced by addition of 0.5% galactose for 4.5 h. 3 $OD_{600}$ were harvested, resuspended in 100 µl medium containing 2% lactate, transferred into a black 96-well plate, and centrifuged (5 min at 30 $g$). The fluorescence was measured with the excitation/emission wavelength 485-15/530-20 nm in a fluorescence microplate reader (Clariostar, BMG Labtech) and by microscopy. The sequence of GFP10 can be found at Addgene, plasmid pSJ2039 (pRS316-NOP1pr-GFP1-10-SCS2TM). This was a gift from Sue Jaspersen (Addgene plasmid # 86418; http://n2t.net/addgene:86418; RRID:Addgene_86418). The sequence of GFP11 was taken from the plasmid pSJ1602 (pRS315-NOP1pr-mCherry-SCS2TM-GFP11).

## Hsf1 activity reporter assay

The cells were grown to log phase ($OD_{600}$ value of 0.4–0.8) in YPGal at 30°C. Part of every culture was incubated for 4 h before the measurement was started at 37°C. Equal amounts of cells (5.0 OD (600 nm)) were collected by centrifugation (4,000 $g$, 5 min, room temperature) and resuspended in 666.6 µl MES-Tris pH 6.8. 200 µl of this cell suspension (1.5 OD (600 nm)) was transferred to a flat-bottomed black 96-well imaging plate (BD Falcon) in technical replicates. Cells were sedimented by gentle spinning (30 × $g$, 5 min, room temperature), and fluorescence (excitation 497 nm, emission 540 nm) was measured using a ClarioStar Fluorescence plate reader (BMG Labtech).

## PROTEOSTAT staining

To assess the occurrence of protein aggregates in cells, the cells were grown in lactate selective media to log phase and then supplemented with 2% galactose to induce the overexpression of Pet9 for 4.5 h. Then, 5.0 OD (600 nm) were harvested and fixed by the addition of formaldehyde to a final concentration of 4% in 1 ml of sterile water for 15 min. After washing two times with PBS, the cells were resuspended in 1 ml PROTEOSTAT (Enzo Life Sciences GmbH) buffer and incubated at room temperature for 30 min. After centrifugation, the cell pellet was resuspended in 250 µl PROTEOSTAT reagent (1:2,000 diluted in PROTEOSTAT buffer) and incubated for 30 min at 4°C. The cells were washed with 500 µl of buffer and then resuspended in 500 µl of buffer. 100 µl of this cell suspension was transferred to a flat-bottomed black 96-well imaging plates in technical replicates. Cells were sedimented by gentle spinning (30 × $g$, 5 min, room temperature), and fluorescence (excitation, 550 nm; emission, 600 nm) was measured using a ClarioStar Fluorescence plate reader (BMG Labtech).

## Co-expression analysis of mitochondrial import components with the cytosolic proteostasis system

To analyze the co-expression of mitochondrial import components with components of the proteostasis network, the Serial Pattern of Expression Levels Locator (SPELL) database (Hibbs, Hess *et al*, 2007) was queried for the set of major cytosolic stress-reactive proteostasis factors (*SSA1, SSA2, SSA3, SSA4, HSP82, HSC82,*

*HSP42, HSP104, HSP26, YDJ1, XDJ1, CCT2, CCT3, PRE1, PRE2, PUP1, RPN1, RPN2, RPT1, RPT2, CDC48)* with the online interface provided by the *Saccharomyces* genome database (SGD, https://spell.yeastgenome.org). All genes in the yeast genome were ranked according to their co-expression score across all transcriptomics datasets deposited to SGD, and mitochondrial import components were highlighted.

### Protein purification and antibody production

To generate an antibody recognizing Rnq1, the *RNQ1* ORF was cloned into the expression plasmid pET19b using NdeI and BamHI restriction sites for the bacterial expression of N-terminal $His_6$-tagged Rnq1. This recombinant protein was expressed in *E. coli* Rosetta cells at 37°C. Once growth reached an OD (600nm) value of 0.45, the cells were subjected to induction by 1 mM IPTG for 2 h. Cells were harvested by centrifugation and lysed in 400 µl buffer L1 (25% sucrose 50mM Tris/HCl pH 8.0) for 10 min. Then, 17.4 ml L1 was added and supplemented with 25 mM EDTA, 1% Triton X-100, 1 mM PMSF, and 10 mM DTT. The lysate was frozen overnight at −70°C. Subsequently, the lysates were sonicated three times for 10 s and centrifuged at 47,000 *g* for 30 min. After resuspending the pellet in 10 ml L2 buffer (20 mM Tris/HCl pH 7.4, 1 mM EDTA, 1 mM PMSF, 1% Triton X-100, 50 mM DTT), another sonication step followed by centrifugation was performed. The resulting pellet was resuspended in 10 ml buffer L3 (L2 + 0.1% Triton X-100), sonified, and centrifuged. Then, the pellet was resuspended in 10 ml buffer L4 (L2 without Triton X-100), sonicated and centrifuged, and afterward resuspended in 1 ml buffer L5 (7 M urea, 50 mM Tris/HCl pH 7.4, 50 mM DTT). After a final sonication step, the lysate was frozen at −20°C until it was used for the immunization of rabbits.

### Antibodies

Antibodies used in this study are listed in Table EV2. Antibodies for the use in *S. cerevisiae* were raised in rabbits using recombinant purified proteins. The secondary antibody was ordered from Bio-Rad (Goat Anti-Rabbit IgG (H + L)-HRP Conjugate). The horseradish peroxidase-coupled HA antibody for Western blotting was obtained from Roche (Anti-HA-Peroxidase, High Affinity (3F10)). Antibodies were diluted in 5% nonfat dry milk-TBS.

### Immunocytochemistry

48 hours after transfection, cells were washed twice in PBS (−/−) and fixed in 4% PFA for 10 min at room temperature. Then, samples were incubated with blocking solution (5% goat serum in 0.2% Triton X-100) for two hours at room temperature. Afterward, the primary antibody was prepared in blocking solution at the indicated concentration and incubated overnight at 4°C in a humidified environment. The next day, coverslips were washed twice in PBS (−/−) and incubated with secondary antibody in PBS (−/−) for 2 h at room temperature. Then, coverslips were washed twice in PBS (−/−) for 10 min, twice in 0.2% Tween in PBS, and twice in PBS (−/−) for 10 min. Finally, cells were stained with DAPI and mounted on coverslips with Fluoromount medium (Thermo Fisher).

### Coating of coverslips

Poly-L-lysine (25 µg/ml in PBS −/−) was applied to autoclaved coverslips and incubated for 1 h. Then, coverslips were washed twice with PBS and laminin solution (4 ng/ml in PBS +/+) was added and incubated overnight in the incubator. The next day, the glass coverslips were washed twice in PBS and cells were seeded.

### Microscopic analysis

To analyze the formation of aggregates in yeast, mid-log phase cultures were shifted to media containing galactose to induce the expression of Q97-GFP and incubated at 30°C for different times. After centrifugation of 1.2 OD (600 nm) of cells (1 min at 16,100 × *g* at room temperature), the cells were washed and resuspended in 250 µl of sterile water. Subsequently, the cell suspension was fixed on a microscope slide for fluorescence imaging using the 100x oil objective of a Nikon Eclipse E600 Microscope. Microscopy images were processed in Fiji.

For fluorescence microscopy and quantification of different aggregation phenotypes at different time points, yeast cells were incubated 1.5–24 h under inducing conditions. 1 ml of cell culture was pelleted, washed, and resuspended in 100 µl of sterile water. To assess the nature of the aggregation pattern in the different strains, 100 cells of each sample were counted and evaluated regarding the aggregation pattern they exhibited.

To assess the cellular localization of Q97-GFP and mitochondrial morphology in the presence of Q97-GFP aggregates in yeast, cells were incubated under non-expressing conditions at 30°C for 16 h. For induction of Q97-GFP expression, cells were then shifted to galactose medium, diluted twofold, and incubated at 30°C for 4 h. Living cells were analyzed using a Leica DMi8 fluorescence microscope (Leica Microsystems GmbH, Wetzlar, Germany) with HC PL APO 100×/1.40 OIL objective, a Lumencor SPECTRA X light source and fluorescence filter sets (FITC ex. 460–500 nm; em. 512–542 nm and TXR ex. 540–580 nm; em. 592–668 nm). The microscope was equipped with a sCMOS Leica-DFC9000GT-VSC07400 camera. Voxel size of the shown images is 0.066 µm. For microscope settings, image generation, and processing (cropping, maximum intensity projection), the Leica LAS X software (version 3.4.2.18368, Leica Microsystems GmbH, Wetzlar, Germany) was used. Deconvolution of *z* stacks was carried out by Huygens Deconvolution Software (Scientific Volume Imaging, Hilversum, The Netherlands). Final image processing, including adjustment of brightness, contrast and background reduction, and the overlay of different channels, was done using Adobe Photoshop CS5 (Adobe Systems).

To perform microfluidics, microfluidic chips were made as previously described (Goulev, Morlot *et al*, 2017; Morlot *et al*, 2019). The microchannels were cast by curing PDMS (Sylgard 184, 10:1 mixing ratio) and then covalently bound to a 24 × 50 mm coverslip using plasma surface activation (Diener, Germany). The assembled chip was then baked for 1 h at 60°C and then perfused with media using Tygon tubing and a peristaltic pump (Ismatec, Switzerland) at a 10 µl/min flow rate. After 2 h of PDMS rehydration, yeast cells were loaded into the chip with a 1-ml syringe. Cells were grown on chip with constant perfusion of glucose selective medium (SD-Leu-Ura) until most cavities were filled with cells (typically overnight). Then induction was started by changing the medium to SC+2%Galactose-

Leu-Ura. At the same time, time-lapse acquisition was initiated. For the glucose recovery experiments, medium was switched back to SD-Leu-Ura after 22-h acquisition of galactose induction. For all other experiments, the same medium (SC+2% Galactose-Leu-Ura) was perfused until the end of the time-lapse. To this end, cells were imaged using an inverted Nikon Ti-E microscope. Fluorescence illumination was achieved using LED light (Lumencor), and emitted light was collected using a 60x N.A. 1.4 objective and a CMOS camera Hamamatsu Orca Flash 4.0. An automated stage was used to follow up to 34 different fields of view in parallel over the course of the experiment. Images were acquired every 10 min for a total duration of 48h using NIS software. Focus was maintained using the Nikon Perfect Focus System. A physiological temperature of 30°C was maintained on the chip using a custom sample holder with thermoelectric modules, an objective heater with heating resistors, and a PID controller (5C7-195, Oven Industries). After acquisition, NIS raw data were analyzed using the custom MATLAB software phylocell and autotrack, available on https://www.github.com/gcharvin. Cell contours and fluorescent aggregates were segmented using a modified watershed algorithm, and tracking was achieved with the Hungarian method, as previously described (Goulev *et al*, 2017; Morlot *et al*, 2019). "Mean fluorescence per cell" corresponds to the average GFP intensity within the cell contours. "Intensity after aggregation" corresponds to the total GFP fluorescence within the segmented aggregate at the first frame where aggregation starts. "Intensity before aggregation" is the total fluorescence within the cell contour in the frame preceding aggregation.

To analyze Q97-GFP and mitochondrial morphology in mammalian cells, fluorescence microscopy was performed using a Zeiss ELYRA PS.1 Super-Resolution Structured Illumination Microscope (SR-SIM) with a 63× oil immersion objective. All channels were acquired independent and subsequently. Single cell analysis was performed in this microscope as well, quantifying only the Htt-Q97 (+/− MIA40-HA) transfected cells. Among the transfected cells, those showing nuclear or cytoplasmic huntingtin aggregates were included in the category "transfected cell with aggregates". The percentage of cells with aggregates was calculated over the total number of transfected cells. Raw confocal SIM images were processed and generated using the Black ZEN and Blue ZEN software.

### Miscellaneous

The following methods were performed according to published methods: isolation of mitochondria and oxygen consumption measurements (Saladi, Boos *et al*, 2020); analysis of mRNA levels by qRT–PCR (Zöller *et al*, 2020); preparation of semi-intact cells and their use for *in vitro* import experiments (Laborenz, Hansen *et al*, 2019); pulse chase assay and alkylation for *in vivo* analysis of Mia40-mediated import (Peleh *et al*, 2016); the analysis of mitochondrial protein import using the Oxa1-Ura3 reporter assay (Hansen *et al*, 2018); and growth and analysis of MIA40-expressing HEK293T cells (Fischer *et al*, 2013; Murschall, Gerhards *et al*, 2020).

## Data availability

The mass spectrometry proteomics data have been deposited to the ProteomeXchange Consortium via the PRIDE (https://www.ebi. ac.uk/pride/) (Perez-Riverol, Csordas *et al*, 2019) partner repository with the dataset identifier PXD023902.

**Expanded View** for this article is available online.

## Acknowledgements

We thank Sabine Knaus, Andrea Trinkaus, and Laura Buchholz for technical assistance, Agnieszka Chacinska for the *mia40-3* mutant stain, Kai Hell for the *GAL*-Mia40 strain, Flaviano Giorgini for the *GALL*-Q103GFP plasmid, Sue Jaspersen for split-GFP plasmids, Katja G. Hansen for the Oxa1-Ura3 plasmid, and Nils Wiedemann and Nikolaus Pfanner for antibodies. This study was funded by grants from the Deutsche Forschungsgemeinschaft (DIP MitoBalance, SPP1710, IRTG1830, HE2803/10-1 to JMH, CRC1218/B02 to JR, WE2174/7-1 to BW, and WI/2111-6 and WI/2111/8 to KFW), the Landesschwerpunkt BioComp (to JMH), the Joachim Herz Stiftung (to FB), and the Elitenetzwerk Bayern (Biological Physics program, to JS). SR-SIM microscopy was funded by the Deutsche Forschungsgemeinschaft and the State Government of North Rhine-Westphalia (INST 213/840-1 FUGG). Open Access funding enabled and organized by Projekt DEAL.

## Author contributions

AMS, KK, TF, and LK designed, cloned, and verified the constructs and strains. JMH conceived the project. AMS, KK, TF, and LK characterized the role of Mia40 in the context of polyQ-induced stress in yeast. SM and GC analyzed the polyQ aggregation by microscopy using microfluidics chambers and live cell imaging. NK and SA programmed the automated detection of aggregation patterns using machine learning. JS, RJB, and BW analyzed the influence of polyQ aggregates and Mia40 on mitochondrial morphology in yeast. ASV, LMM, JR, and KFW planned and carried out the experiments in human cells to analyze the relevance of MIA40 in the context of polyQ aggregation. PH, FS, FB, and AMS analyzed the cellular proteome by mass spectrometry. All authors analyzed the data. JMH wrote the manuscript with assistance from all other authors.

## Conflict of interest

The authors declare that they have no conflict of interest.

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
