## [Review Process File · The EMBO Journal]

Increased levels of mitochondrial import factor Mia40 prevent the aggregation of polyQ proteins in the cytosol

Anna Schlagowski, Katharina Knöringer, Sandrine Morlot, Ana Sanchez-Vicente, Tamara Flohr, Lena Kraemer, Felix Boos, Nabeel Khalid, Sheraz Ahmed, Jana Schramm, Lena Maria Murschall, Per Haberkant, Frank Stein, Jan Riemer, Benedikt Westermann, Ralf Braun, Konstanze Winklhofer, Gilles Charvin, and Johannes Herrmann

DOI: [10.15252/embj.2021107913](https://doi.org/10.15252/embj.2021107913)

Corresponding author: Johannes Herrmann (hannes.herrmann@biologie.uni-kl.de)

Review Timeline:

Submission Date:	2nd Feb 21
Editorial Decision:	5th Mar 21
Revision Received:	28th Apr 21
Editorial Decision:	18th May 21
Revision Received:	25th May 21
Accepted:	31st May 21

Editor: Elisabetta Argenzio

Transaction Report:

Thank you for submitting your manuscript entitled "Increased levels of the mitochondrial import factor Mia40 prevent the aggregation of polyQ proteins in the cytosol" [EMBOJ-2021-107913] to The EMBO Journal. Your study has now been assessed by three reviewers, whose reports are enclosed below for your information.

As you can see, the referees concur with us on the general interest of your findings but also raise several issues that need to be solved before they can support publication in The EMBO Journal. In particular, they ask you to test whether other mitochondrial import pathways are involved in protecting cells from polyQ cytotoxicity (referee #1 and #2), as well as the effects of overexpression of a mitochondrial precursor protein in your system (referee #2).

Given the overall interest of your study, we have decided to invite you to submit a new version of the manuscript revised according to the referees' requests. I should add that it is The EMBO Journal policy to allow only a single round of revision, and acceptance of your manuscript will therefore depend on the completeness of your responses in the revised version, as well as unanimous strong support by the reviewers.

I realize that addressing all the referees' criticisms could be not only time-consuming and labour-intensive but also technically challenging. Therefore, I would be available to discuss the revisions further either via video or phone - please let me know what works best for you.

We generally grant three months as standard revision time. As we are aware that many laboratories cannot function at full capacity owing to the COVID-19 pandemic, we may relax this deadline. Also, we have decided to apply our 'scooping protection policy' to the time span required for you to fully revise your manuscript and address the experimental issues highlighted herein. Nevertheless, please inform us as soon as a paper with related content is published elsewhere.

When preparing your letter of response to the referees' comments, please bear in mind that this will form part of the Review Process File and will therefore be made available online. For more details on our Transparent Editorial Process, please visit our website:
http://emboj.embopress.org/about#Transparent_Process

Before submitting your revised manuscript, deposit any primary datasets and computer code produced in this study in an appropriate public database (see <http://msb.embopress.org/authorguide#dataavailability>). Please remember to provide a reviewer password, in case such datasets are not yet public. The accession numbers and database names should be listed in a formal "Data Availability" section (placed after Materials & Method). Provide a "Data availability" section even if there are no primary datasets produced in the study.

I thank you again for the opportunity to consider this work for publication and look forward to your revision.

Referee #1:

Our understanding of mitochondrial functions is being advanced in such a way that the failure of mitochondrial protein import tends to give a profound impact onto other organelle and cytosolic proteostasis. This study by Schlagowski et al. adds another novel and important findings to this. Mitochondrial protein import relies on several different pathways, but the authors indicated that MIA pathway or Mia40 plays a rate-limiting role for mitochondrial protein import, thereby contributing to the maintenance of the cytosolic proteostasis. Depletion of Mia40 makes the cell hypersensitive to a polyQ protein, Q97-GFP, and in turn overexpression of Mia40 suppresses the formation of toxic Q97-GFP aggregates. The authors presented several pieces of evidence that the compromised Mia40 function leads to cytosolic accumulation of its substrates, which results in failure of preventing seed formation of Rnq1 and the following large aggregate formation of polyQ proteins. Large aggregate formation of polyQ proteins then competes with the import of mitochondrial proteins other than the MIA substrates. I have the following concerns or questions to be clarified by the authors.

(1) Although the authors discussed the point, I would still like to have the following points clarified or at least considered for further experimental assessment. A major question is why Mia40? Although the authors appear to think negatively about such a possibility, this could be due to the properties of MIA substrates accumulated in the cytosol. The authors can test this possibility simply by expressing import-incompetent MIA substrate derivatives, such as MIA substrates fused to a long, but not aggregate-prone polypeptide chain with co-expression of polyQ proteins. Alternatively, if this is due to the compromised import of mitochondrial proteins other than MIA substrates as the authors considered in Discussion, the authors can expect a similar effect on polyQ proteins by suppressing the TIM22 or SAM pathways that require MIA substrates, i.e. small TIM proteins.

(2) What is the difference between the small polyQ aggregates and large poly Q aggregates generated by overexpressed Mia40? The authors performed MS analysis of the soluble fraction after induction of Mia40, but what about the pellet fraction? If MS analysis is difficult to apply for the pellet fraction, immunodetection can be performed alternatively.

(3) The chronological and causative sequence of the followings is not clearly demonstrated and/or discussed in the manuscript; compromised Mia40 function, accumulation of MIA substrates, the Rnq1 seed formation, the formation of large aggregates involving polyQ proteins, and inhibition of the import of mitochondrial proteins other than MIA substrates. At least, the sequence of these events could be discussed more clearly.

(4) The effects of the compromised Mia40 function on its substrates and other mitochondrial proteins together with co-expression of polyQ proteins had better be assessed in a quantitative manner. The authors can quantify the amounts of MIA substrates and other mitochondrial proteins that accumulated in the cytosol, and then compare them with those in wild type cells.

Minor point - In page 14, line 5, Fig. 6G should read Fig. 6F.

Referee #2:

The authors present an interesting study on the crosstalk of mitochondrial protein biogenesis and cytosolic proteostasis, a quite novel and highly exciting research field. One of the central findings is that aggregation of 'longQ' variants of polyQ proteins in the cytosol (expressed in the model yeast)

can be prevented by increasing levels of the mitochondrial intermembrane space import component Mia40. Impairment of Mia40 leads in turn to increased proteotoxicity. These findings are further accompanied by an array of data including proteomics and transfer of their findings into a human cell system (SH-SY5Y).

This is a very interesting observation and the experimental approach is clear and the presented data are of high quality. The paper will not only be of high relevance for this new but already quite competitive research field but will also be interesting to a broader audience. However, I still have several issues that should be clarified before publication.

- it is not quite clear to me why the authors used polyQ proteins in their assays, which is a bit an artificial system. Given that PolyQ proteins and mitochondrial precursor proteins accumulating in the cytosol add up to proteotoxicity, why can't one simply overexpress a mitochondrial precursor protein? This might reflect a more physiological scenario, particularly given, that it was recently shown that non-processed mitochondrial precursor proteins are indeed prone to aggregation (see Poveda-Huertes et al., Molecular Cell (2020)). Did the authors tested this (and the respective role of Mia40)?

- Did the authors test potential involvement of other mitochondrial import pathways?

Particularly, in Fig 6B they test import of Hsp60 which takes the presequence import pathway. What about the import efficiency of Mia40 substrates here (such like the small Tim proteins...)? One could here also test for accumulation of non-processed precursor in the cytosol. Wouldn't this be a more direct approach than the Oxa1-Ura3 assay?

Was the involvement of any other import machinery component tested for protective or proteotoxic effects when overexpressed or mutated upon PolyQ expression?

What about the further components of the MIA pathway? Would e.g. overexpression of Erv1 have the same effect as Mia40?

- The Rnq1 link appears very interesting. Did the authors test a potential involvement of other players mentioned here, such like Hsp104?

- Fig. 2E: How do the authors know if the cells undergo cell death and are not just entering G0 phase?

- Page 12 bottom/Fig 6C: after 4.5 h induction I would not expect a significant change of protein levels, particularly of carrier proteins, because of the quite long half lifes of mitochondrial proteins. Was a longer induction time tested?

- is there any idea/model how Mia40 in the end would exert its protective role against cytosolic protein aggregates?

Referee #3:

- general summary and opinion about the principle significance of the study, its questions and findings

The purpose of this study is to elucidate the role of Mia40 in stabilizing cytosolic proteostasis. The authors show that the mia40 mutants are more susceptible to polyQ toxicity than wild-type cells and Mia40 overexpression suppresses polyQ aggregation and toxicity in yeast. The authors

suggest that Mia40 overexpression increases the resistance against polyQ toxicity by alteration of Rnq1 levels. In addition, they show that cytosolic polyQ aggregates interfere with the efficient protein import into mitochondria. The authors confirm a protective effect of Mia40 against polyQ toxicity also in human cells. Although this is an interesting study demonstrating that Mia40 reduces a negative effect of Q97 aggregation, there are many critical questions that need to be addressed to support the conclusions made by the authors.

- specific major concerns essential to be addressed to support the conclusions

1) Figure 1 : the authors show that Q97-GFP is aggregated in the cytosol. However, this is already reported by other groups (Gruber, Hornburg et al., 2018, Klaips, Gropp et al., 2020), this figure is more appropriated to be presented as an extended data figure and the previous papers should be cited commenting on the agreement with their results.

Gruber A, Hornburg D, Antonin M, Krahmer N, Collado J, Schaffer M, Zubaite G, Luchtenborg C, Sachsenheimer T, Brugger B, Mann M, Baumeister W, Hartl FU, Hipp MS, Fernandez-Busnadiego R (2018) Molecular and structural architecture of polyQ aggregates in yeast. *Proc Natl Acad Sci U S A* 115: E3446-E3453

Klaips CL, Gropp MHM, Hipp MS, Hartl FU (2020) Sis1 potentiates the stress response to protein aggregation and elevated temperature. *Nat Commun* 11: 6271

2) Figure 1E : These results do not correspond to images of Q97-GFP after 24h induction in Figure 1F. The big aggregates are present in the microscopy images, but the majority of Q97-GFP is present in the soluble fraction (gels). Moreover, after 6h there are more insoluble aggregates than after 24h. The authors should explain where this difference is coming from; do authors suggest that big aggregates are more soluble or gone? Authors should also examine the pellet fraction after cell lysis to show that aggregates are not present there.

3) Figure 1F, G : the aggregates increase to almost 100% after 6h of induction in the figure 1G. However, less than 50% of cells contains protein aggregates in the figure 1F image. The image should be replaced by a representative one showing 100% of aggregation, or the number of cells with aggregates in the figure 1G should be re-counted.

4) Figure 2A : the authors show that mitochondria and aggregates are not co-localized. However, the co-localization of them is clearly visible (yellow color). To make it clear, the authors should provide not only merged images but also individual images of mtmCherry and Q97-GFP, and magnified images. Providing the Pearson's co-localization coefficient will bring accuracy to this co-localization analysis. In addition, subcellular fractionation may be performed to support the conclusions drawn from the data of microscopic analysis.

5) Figure 2D : As the mutant used in this study is temperature sensitive, the authors should present the additional data of drop dilution experiments performed in low and high temperatures (19{degree sign}C and 37{degree sign}C). The 25{degree sign}C is in the middle of the defect stimulation.

6) Figure 3B: the authors show that Q97-GFP forms several small aggregates in WT cells, whereas it forms a large aggregate in the GAL-Mia40 overexpression strain. In the same line with the comment for the figure 1E, the authors should examine the cell debris pellet to test if there are large aggregates. Q97-GFP forms a very large aggregate in the GAL-Mia40 as shown in the figure 3C, they may be found in the discarded pellet.

Soluble Q97-GFP (~60kD) in this figure is more abundant than that of 6h after induction in the figure 1E in wild-type cells. The authors are asked to explain where the difference is coming from.

7) Figure 3C : The aggregation distribution of Q97-GFP in wild-type cells in this figure is different from the figure EV1. In the figure EV1, after 4 h, there are bigger dots of Q97 aggregates in WT than in GAL-Mia40 strains, what is in contrary to the author's interpretation in the figure 3C. The authors should explain what makes the difference.

Aggregation mode of Q97 in the presence of GAL-Mia40 in each cell can depend on the expression

level of each protein. This might justify the differences presented in the figure 3D. Therefore, the authors are asked to also visualize Mia40 in the microscopy images and provide data that will present levels of Mia40 and Q97 in each cell at the same time, only then the conclusions can be driven.

The authors should comment on the change of the Q25-GFP signal distribution in GAL-Mia40/WT presented in the figure.

8) Figure 3D : The authors quantified the Q97-GFP patterns of aggregates. However, it is unclear what parameters authors used to define which one is small and which one is big. Big dots are visible 6h after induction in WT in the figure 1F, but they were not counted as big dots in the figure 3D (upper graph). The authors should explain how the dots were quantified and why small multiple dots appear 12h after induction in GAL-Mia40 strain.

9) Figure 5B, E : There is less Rnq1 in the soluble fraction of GAL-Mia40 strain compared to that of WT 4,5h after Q97 induction in mass spectrometry data (Fig 5B). However, Rnq1 is more abundant in the soluble fraction of GAL-Mia40 strain than in that of WT (Fig 5E). The authors should explain the discrepancy.

10) Figure 5E, F : Rnq1 is insoluble in the GAL-Mia40(N), whereas it is soluble in the previous GAL-Mia40 strain. The authors should show whether the GAL-Mia40(N) reproduces the data in the figure 3B, C and D. The authors should examine the Mia40 expression and Q97-GFP aggregation in the GAL-Mia40(N). Additionally, the authors should explain how they exclude the possibility that effect of GAL-Mia40 adaptation doesn't stimulate the secondary effect which are actually responsible for the change of Q97 aggregation. The authors should include such discussion in the manuscript.

11) Figure 6B : the same experiment should be done with GAL-Mia40+empty as a control of GAL-Mia40+Q97. The quality of the western blot data is not satisfactory and it does not allow to draw any conclusion. If not improved the data and conclusions drawn from it should be removed from the manuscript.

12) Figure 6C : in the mass spectrometry data of Table EV1, TOM20, TOM22, TOM70 are also significantly increased in Q97 expressed cells. However the authors focus only on Mia40 as the factor main factor. The authors should check whether any of TOM proteins have the same effect as Mia40.

Mia40 upregulation is around 25% due to Q97 expression. The authors show the Mia40 overexpression that significantly exceeding 25% prevents Q97 aggregation. Therefore this is artificial experimental setup. The authors should extend discussion how relevant it is in physiological condition and what could stimulate such big Mia40 response in cells.

13) Figure 6E : the authors claim that elevated Mia40 is beneficial under stress conditions showing GAL-Mia40 strain is less reactive to heat stress. HSF1 activation is generally thought to be required for protection from proteotoxicity, so it is questionable whether less HSF1 activation is indeed beneficial to cells. The authors should provide evidence that GAL-Mia40 strain has less proteotoxicity under heat shock stress than WT. (e.g., PROTEOSTAT staining of WT and GAL-Mia40 strains after heat shock). Before drawing conclusions, the authors should also include statistical analysis in this figure 6E, and also include the data of GAL-Mia40+Q97.

14) Figure 7B, D, EV4A : MIA40-HA seems to be fully merged with Q97-GFP in human cells. Does this data indicate MIA40 directly interact with Q7-GFP?

15) Figure 7E : The summary figure should be focused on the data presented in the paper. There are not enough evidence presented in the manuscript to make such summary.

16) page 12, last paragraph : the authors state that mitochondrial carrier protein levels are decreased with Q97 expression. To complete the statement, the authors should examine if this is just specific response to carrier proteins.

- minor changes to be addressed/corrected

1) The method of western blotting analysis is missing

- 2) The method of statistical analysis is missing
- 3) standard deviations/errors of the mean are missing in the figure 2B and C
- 4) Figure 3B : the lower exposure blot should be added because based on the data, the level of Q25 in GAL-Mia40 seems to be increased when compared to wild-type.
- 5) Figure 4D : indicate what 'M' and 'D' mean in the figure legend.
- 6) Figure 5G : the authors should present the drop dilution experiment data of the WT(N) in the figure.
- 7) Figure 6A : the authors should support data with additional figures of the CHX chase assay showing cytosolic and mitochondrial Mia40 protein levels within the frame of experiment.
- 8) Figure 6G : this data should be introduced in more details for non-expert readers.
- 9) Figure 7B, D : the fluorescence intensity of TOM20 and TRAP1 should be increased. They are hardly visible in Δ N-MIA40 expressed cells.
- 10) page 3, line 12 : use of "Gruber, Hornburg et al., 2018" citation is not appropriate. The authors should differentiate the aggregates association with membranes in yeast and mammals.
- 11) end of page 4 : the authors should specify whether the relation of Mia40 and Rnq1 is based on their result, or an appropriate citation is needed.
- 12) page 5, first sentence in the result section: the authors should appreciate Susan Lindquist's contribution in the field.
- 13) page 5, end of first paragraph : the authors should select a better citation. "Duennwald et al., 2006" is based on Q25 and Q103 not Q97.
- 14) end of page 6 : the authors should explain how Mia40 was selected for this study.
- 15) page 9 : "these numbers" -The authors should change it to more precise language.
- 16) page 10, end of second paragraph. There is no information to which figure this result section corresponds to. Reference to Figure 5B should be added in an appropriate place.
- 17) top of page 12 : The authors should provide a citation for the statement of no indication of Mia40 presence in extra-mitochondrial fractions or perform the fractionation experiment supporting this statement.
- 18) page 14, line 5 : "Fig. 6G" should be replaced by "Fig. 6F"
- 19) page 15, line 11 : "EV4B" should be replaced by "EV4A"

Specific points of the referees

Referee #1:

Our understanding of mitochondrial functions is being advanced in such a way that the failure of mitochondrial protein import tends to give a profound impact onto other organelle and cytosolic proteostasis. This study by Schlagowski et al. adds another novel and important findings to this. Mitochondrial protein import relies on several different pathways, but the authors indicated that MIA pathway or Mia40 plays a rate-limiting role for mitochondrial protein import, thereby contributing to the maintenance of the cytosolic proteostasis. Depletion of Mia40 makes the cell hypersensitive to a polyQ protein, Q97-GFP, and in turn overexpression of Mia40 suppresses the formation of toxic Q97-GFP aggregates. The authors presented several pieces of evidence that the compromised Mia40 function leads to cytosolic accumulation of its substrates, which results in failure of preventing seed formation of Rnq1 and the following large aggregate formation of polyQ proteins. Large aggregate formation of polyQ proteins then competes with the import of mitochondrial proteins other than the MIA substrates. I have the following concerns or questions to be clarified by the authors.

We thank referee 1 for her/his very positive evaluation of our study.

(1) Although the authors discussed the point, I would still like to have the following points clarified or at least considered for further experimental assessment. A major question is why Mia40? Although the authors appear to think negatively about such a possibility, this could be due to the properties of MIA substrates accumulated in the cytosol. The authors can test this possibility simply by expressing import-incompetent MIA substrate derivatives, such as MIA substrates fused to a long, but not aggregate-prone polypeptide chain with co-expression of polyQ proteins. Alternatively, if this is due to the compromised import of mitochondrial proteins other than MIA substrates as the authors considered in Discussion, the authors can expect a similar effect on polyQ proteins by suppressing the TIM22 or SAM pathways that require MIA substrates, i.e. small TIM proteins.

We followed the suggestion of the referee and generated three data sets which we now add to the study: (1) We tested the toxicity of Q97-GFP in two other temperature-sensitive import mutants. In consistence with what we had observed with temperature-sensitive *mia40-3* cells, Q97-GFP induction increased the lethality also in temperature-sensitive *erv1* and *tom40* mutants (see Figure 1). (2) Next, we tested whether overexpression of other components of the mitochondrial import machinery also have the potential to suppress the toxicity of Q97-GFP. Strikingly, we observed that the overexpression of Tim9, Erv1, Sam37 and Sam50 indeed suppressed polyQ toxicity. This suppressive effect was very strong and comparable to the well-established suppression by the disaggregase Hsp104. As this referee suggests, this points to a particular relevance of the MIA/TIM22/SAM pathways. In consistence, we observed no suppression by overexpression of TOM/TIM23 components. (3) We also followed the suggestion to study more explicitly the fate of MIA substrates. To this end, we developed a split GFP assay to monitor the interaction of Erv1 with the cytosolic Hsp70 Ssa1. Upon expression of Q97-GFP we observed co-localization of Erv1 with Ssa1 (novel Fig. 6D, E) in cytosolic puncta, suggesting that polyQ aggregates impair the import of MIA substrates. This is an exciting observation and further studies will have to elucidate in more depth which import pathways are affected by cytosolic aggregates and how.

(2) What is the difference between the small polyQ aggregates and large poly Q aggregates generated by overexpressed Mia40? The authors performed MS analysis of the soluble fraction after induction of Mia40, but what about the pellet fraction? If MS analysis is difficult to apply for the pellet fraction, immunodetection can be performed alternatively.

PolyQ aggregates form very dense structures which even are insoluble to SDS and other denaturing conditions. This is why we performed the quantitative mass spectrometry exclusively on the NP40-soluble fraction. Inspired by the suggestion of the referee, we now developed a protocol to purify Q25/97-GFP by affinity chromatography using sepharose-coupled GFP-specific nanobodies. We performed mass spectrometry using a quantitative TMT-labeling strategy on three independent replicates for each sample. We identified 1000 proteins from which about 400 could be quantified. Unfortunately, the result clearly showed, that Q25-GFP from wild type cells and Q97-GFP from *GAL-Mia40* cells were efficiently recovered, but only very low levels of Q97-GFP were detected in wild type samples (Fig. 1). This clearly demonstrated that the insolubility of the polyQ aggregates strongly biased the proteomic analysis. We therefore decided not to include the data.

Fig. 2. Proteomics of interactors of Q25/97-GFP. Shown are the levels of identified Q25/97-GFP 4.5 h after induction in wild type and *GAL-Mia40* cells. These measurements demonstrate the technical problems in the purification or detection of the Q97-GFP protein from wild type samples.

(3) The chronological and causative sequence of the followings is not clearly demonstrated and/or discussed in the manuscript; compromised Mia40 function, accumulation of MIA substrates, the Rsq1 seed formation, the formation of large aggregates involving polyQ proteins, and inhibition of the import of mitochondrial proteins other than MIA substrates. At least, the sequence of these events could be discussed more clearly.

We now considerably extended the discussion (and the data on this aspect). Our study clearly shows that Q97-GFP leads to an accumulation of mitochondrial proteins in the cytosol, however, there is no strong import block comparable to that of 'clogger' proteins. At least under the conditions of this study, in which we focus on the induction of high levels of Q97-GFP for rather short time, the mitochondria appear to stay largely functional and the toxic effects therefore arise from problems in the cytosol. The sequence of reactions is now better discussed and also better illustrated in the hypothetical model in our study.

(4) The effects of the compromised Mia40 function on its substrates and other mitochondrial proteins together with co-expression of polyQ proteins had better be assessed in a quantitative manner. The authors can quantify the amounts of MIA substrates and other mitochondrial proteins that accumulated in the cytosol, and then compare them with those in wild type cells.

We now added a novel experiment (Fig. 6B) which shows that in Western blots of whole cell extracts, precursor forms of mitochondrial proteins are not detected after Q97-GFP expression for 4.5 h, whereas expression of the 'clogger' cytochrome b₂-DHFR leads to an accumulation of non-imported precursors. Nevertheless, more sensitive methods such as those using Oxa1-Ura3 clearly show the accumulation of cytosolic precursor forms upon Q97-GFP expression. In addition, we now developed a novel assay relying on split superfolder GFP which confirms the cytosolic accumulation of Oxa1 and of the Mia40 substrate Erv1. These data are now shown in the novel Figs. 6D and E.

Minor point - In page 14, line 5, Fig. 6G should read Fig. 6F.

We corrected this

Referee #2:

The authors present an interesting study on the crosstalk of mitochondrial protein biogenesis and cytosolic proteostasis, a quite novel and highly exciting research field. One of the central findings is that aggregation of 'longQ' variants of polyQ proteins in the cytosol (expressed in the model yeast) can be prevented by increasing levels of the mitochondrial intermembrane space import component Mia40. Impairment of Mia40 leads in turn to increased proteotoxicity. These findings are further accompanied by an array of data including proteomics and transfer of their findings into a human cell system (SH-SY5Y). This is a very interesting observation and the experimental approach is clear and the presented data are of high quality. The paper will not only be of high relevance for this new but already quite competitive research field but will also be interesting to a broader audience. However, I still have several issues that should be clarified before publication.

We thank this referee for these very positive statements about the relevance of our observations and the quality of our study.

1. - it is not quite clear to me why the authors used polyQ proteins in their assays, which is a bit an artificial system. Given that PolyQ proteins and mitochondrial precursor proteins accumulating in the cytosol add up

to proteotoxicity, why can't one simply overexpress a mitochondrial precursor protein? This might reflect a more physiological scenario, particularly given, that it was recently shown that non-processed mitochondrial precursor proteins are indeed prone to aggregation (see Poveda-Huertes et al., *Molecular Cell* (2020)). Did the authors test this (and the respective role of Mia40)?

A number of recent studies showed that the cytosolic accumulation of precursor proteins can lead to growth arrest and reduced cellular fitness. These studies mainly used mutants in import components such as Mia40/Tim17 (Wrobel ...Chacinska, 2015) or MPP (Poveda-Huertes ... Vögtle, 2020) or 'cloggers' (Weidberg, Amon, 2018; Boos...Herrmann, 2019). Our present study shows for the first time that the inverse correlation might also be true: Improved mitochondrial import can stabilize cytosolic proteostasis. The Poveda-Huertes study mentioned by the referee is exciting, however, in that case the aggregation problem occurs presumably mainly within mitochondria due to the defective processing peptidase. We cite this study now. Furthermore, we added several new experiments to test whether the overexpression of Mia40 reduces the cell growth arrest induced by cloggers (novel Fig. EV4F) or overexpressed mitochondrial carrier proteins (novel Fig. EV4D, E). As expected, these overexpressed proteins remain to be toxic as they obviously still overwhelm the import capacity.

2. - Did the authors test potential involvement of other mitochondrial import pathways? Particularly, in Fig 6B they test import of Hsp60 which takes the presequence import pathway. What about the import efficiency of Mia40 substrates here (such like the small Tim proteins...)? One could here also test for accumulation of non-processed precursor in the cytosol. Wouldn't this be a more direct approach than the Oxa1-Ura3 assay? Was the involvement of any other import machinery component tested for protective or proteotoxic effects when overexpressed or mutated upon PolyQ expression? What about the further components of the MIA pathway? Would e.g. overexpression of Erv1 have the same effect as Mia40?

We followed the excellent suggestions of the referee and generated two novel sets of experiments: (1) we tested whether the overexpression of other factors of the mitochondrial import machinery also suppress polyQ toxicity. As shown in novel Figs. 7B and EV4C, overexpression of Erv1, Tim9, Sam50 or Sam37 also suppressed the Q97-GFP-induced growth arrest. This indeed points to a specific role of the import pathways of mitochondrial membrane proteins that are made without presequence. See also our comment to point 1 of referee 1. (2) We performed two additional experiments for detection of cytosolic precursors. First, we followed the suggestion of referee 2 and detected precursor forms of Mdj1 and Rip1 by Western blotting. Consistent with our model, expression of polyQ proteins did not lead to their accumulation, in contrast to conditions where an import clogger was expressed. Second, we used an in vivo approach with split GFP, which showed an increased co-localization of Erv1 and Oxa1 with cytosolic Ssa1 in the presence of Q97-GFP.

We are grateful for the suggestion of these sets of experiments, which clearly improved our study.

3. - The Rnq1 link appears very interesting. Did the authors test a potential involvement of other players mentioned here, such like Hsp104?

Previous studies showed that Hsp104 is necessary for polyQ toxicity, but on the other hand, overexpression of Hsp104 also suppresses polyQ toxicity (e.g. Krobitsch,

Lindquist, 2000). We added a number of experiments which clearly confirmed these previous findings. Novel Figs. EV4A and B show that deletion of Hsp104 in wild type and in the *GAL-Mia40* strain abrogated the Q97-GFP-induced growth difference of both strains. Moreover, in the novel Fig. 7B and EV4C, we show that the overexpression of Hsp104 suppresses polyQ toxicity. These data are consistent with a central role of the Hsp104 chaperone in the control of Q97-GFP/Rnq1 aggregation, that is apparently influenced by mitochondrial precursor proteins and Mia40 overexpression.

4. - Fig. 2E: How do the authors know if the cells undergo cell death and are not just entering G0 phase?

Even if Q97-GFP is only temporarily expressed and again repressed after 4 or 6 h, the cells can never recover. They are dead. This is shown in Figs. 2C and 3E. Moreover, the death of cells is also evident from bursting cells seen in the videos (in the wild type upon Q97-GFP expression, EV_Movie_3-WT.avi).

5. - Page 12 bottom/Fig 6C: after 4.5 h induction I would not expect a significant change of protein levels, particularly of carrier proteins, because of the quite long half lives of mitochondrial proteins. Was a longer induction time tested?

We optimized the induction time in order to minimize secondary effects. To this end, we for example performed initially mass spec experiments at 0, 4.5 and 18 h. As shown here in Fig. 2, the correlation of the changes observed after 4.5 and 18 h were very similar, albeit more pronounced after 18 h. Since we observed that cells started to die after about 5 to 6 h of Q97-GFP induction in wild type cells, we decided to focus on the 4.5 h induction.

Fig. 3. Proteomic analysis of the NP40-soluble proteins in the strains used in this study. Q25-GFP and Q97-GFP were induced for 0, 4.5 and 18 h as described for the experiment shown in Figs. 5B and 7A of our manuscript. This data originates from one of the pilot experiments that we performed to identify a best suited timepoint for our analysis. After 4.5 h of induction, we saw basically the same overall effects as at 18 h, however, most cells were still alive and able to recover once the polyQ induction was stopped.

5. - is there any idea/model how Mia40 in the end would exert its protective role against cytosolic protein aggregates?

Our results are compatible with a central role of precursors of mitochondrial membrane

proteins for protein stability in the cytosol. We propose two mutually not exclusive mechanisms: (1) An increased translocation of precursor proteins, in particular of carriers and other membrane proteins, apparently reduces the occupancy of cytosolic chaperones. (2) The overexpression of specific import factors might also trigger an anti-stress-response which provides additional stress resistance. In both contexts, factors such as Rnq1 and Hsp104 seem to be critical. We now extended the respective sections in the Discussion and improved the model shown in Fig. 8E.

Referee #3:

- general summary and opinion about the principle significance of the study, its questions and findings

The purpose of this study is to elucidate the role of Mia40 in stabilizing cytosolic proteostasis. The authors show that the mia40 mutants are more susceptible to polyQ toxicity than wild-type cells and Mia40 overexpression suppresses polyQ aggregation and toxicity in yeast. The authors suggest that Mia40 overexpression increases the resistance against polyQ toxicity by alteration of Rnq1 levels. In addition, they show that cytosolic polyQ aggregates interfere with the efficient protein import into mitochondria. The authors confirm a protective effect of Mia40 against polyQ toxicity also in human cells. Although this is an interesting study demonstrating that Mia40 reduces a negative effect of Q97 aggregation, there are many critical questions that need to be addressed to support the conclusions made by the authors.

We thank the referee for his/her positive comments and for the many very specific comments. These suggestions were very helpful to improve the manuscript.

- specific major concerns essential to be addressed to support the conclusions

1) Figure 1 : the authors show that Q97-GFP is aggregated in the cytosol. However, this is already reported by other groups (Gruber, Hornburg et al., 2018, Klaips, Gropp et al., 2020), this figure is more appropriated to be presented as an extended data figure and the previous papers should be cited commenting on the agreement with their results. Gruber A, Hornburg D, Antonin M, Kraemer N, Collado J, Schaffer M, Zubaite G, Luchtenborg C, Sachsenheimer T, Brugger B, Mann M, Baumeister W, Hartl FU, Hipp MS, Fernandez-Busnadiego R (2018) Molecular and structural architecture of polyQ aggregates in yeast. Proc Natl Acad Sci U S A 115: E3446-E3453; Klaips CL, Gropp MHM, Hipp MS, Hartl FU (2020) Sis1 potentiates the stress response to protein aggregation and elevated temperature. Nat Commun 11: 6271

We actually chose to use Q25/97-GFP as model proteins in our study because the aggregation behavior of this protein is very well studied in yeast and human cells and was used by many researchers in the past. We are aware of the studies mentioned by the referee (and actually cited and discussed both of these beautiful studies already in the initial version) and the pioneering work of Susan Lindquist's lab and others. However, to our knowledge, all previous studies used growth conditions at which cells grew by fermentation (glucose or galactose as carbon sources). Since Fig. 1 addresses mitochondrial functionality, we used lactate-based media in which polyQ expression was induced by addition of galactose. For this reason, and to introduce the system to our readers, we decided not to move Fig. 1A-C to the supplement. However, we explain this reasoning now better in the text and, as before, make very clear that these model proteins have been used by others before, thereby also acknowledging the contributions from Martinsried.

2) Figure 1E : These results do not correspond to images of Q97-GFP after 24h induction in Figure 1F. The big aggregates are present in the microscopy images, but the majority of Q97-GFP is present in the soluble fraction (gels). Moreover, after 6h there are more insoluble aggregates than after 24h. The authors should explain where this difference is coming from; do authors suggest that big aggregates are more soluble or gone? Authors should also examine the pellet fraction after cell lysis to show that aggregates are not present there.

We agree with the referee that not all of the aggregated species might be detected on gels and by Western blotting. We therefore now added the statement to the text: 'The amounts of aggregated protein are presumably underestimated due to the limited solubility of this protein species.'

3) Figure 1F, G : the aggregates increase to almost 100% after 6h of induction in the figure 1G. However, less than 50% of cells contains protein aggregates in the figure 1F image. The image should be replaced by a representative one showing 100% of aggregation, or the number of cells with aggregates in the figure 1G should be re-counted.

We only counted cells in which GFP was expressed and which therefore showed fluorescence.

The apparent fluorescence intensity of the large aggregates and the distributed smaller aggregates or soluble GFP distributed in the cytosol is very different (aggregates are much brighter), particular at short induction times. This is why in the picture the dispersed signals are hard to see. We show a representative image for inspection of the referee in which the dispersed staining is obvious, using two different settings for the contrast and signal intensity.

Fig. 3. Q97-GFP was expressed in wild type cells for 4 h. An image was generated under the conditions used for Fig. 1G. Note that the dispersed staining is only visible at the high intensity / low contrast conditions of the panel on the right.

4) Figure 2A : the authors show that mitochondria and aggregates are not co-localized. However, the co-localization of them is clearly visible (yellow color). To make it clear, the authors should provide not only merged images but also individual images of mtmCherry and Q97-GFP, and magnified images. Providing the Pearson's co-localization coefficient will bring accuracy to this co-localization analysis. In addition, subcellular fractionation may be performed to support the conclusions drawn from the data of microscopic analysis.

The figure shows maximum intensity projections of z stacks subjected to deconvolution. Therefore, at least in some instances, fluorescent signals might overlap in the projection

even though they are clearly separated in the z axis. As requested by the reviewer, we now provide with this rebuttal letter individual images of mt-mCherry and Q97-GFP at the end of this letter. We show individual optical planes (for the sake of space we show only every second plane). These images were not subjected to deconvolution and therefore are more blurry than the images shown in Fig. 2A.

Please note that the size of the structures that we analyzed are at the limit of resolution of our epi-fluorescence microscopy system (mitochondrial diameter) or even smaller (Q97-GFP aggregates). Therefore, we feel that it doesn't make sense to provide magnified images.

In sum, we feel that the statement we made in the manuscript text accurately reflects our observations: "We observed that mitochondria and aggregates were clearly distinct structures. Mitochondrial colocalization that was seen for a few puncta might represent random contacts." As we don't have any evidence for a specific co-localization of Q97-GFP with mitochondria we feel that further analysis of this question would be beyond the scope of the manuscript and very likely would yield negative results.

5) Figure 2D: As the mutant used in this study is temperature sensitive, the authors should present the additional data of drop dilution experiments performed in low and high temperatures (19°C and 37°C). The 25°C is in the middle of the defect stimulation.

This mutant was not generated by us. It was first described by Chacinska, Pfanner and co-workers (Chacinska,... Pfanner. 2004. EMBO J. 23, 3735f) and the requested drop dilution experiment was already shown in this initial study. Since our study is already extremely packed with data, we decided not to include a further figure which just would confirm already published information. However, in the text we mention that the mutant grows is unaffected at the permissive temperatures used in the experiment.

6) Figure 3B: the authors show that Q97-GFP forms several small aggregates in WT cells, whereas it forms a large aggregate in the GAL-Mia40 overexpression strain. In the same line with the comment for the figure 1E, the authors should examine the cell debris pellet to test if there are large aggregates. Q97-GFP forms a very large aggregate in the GAL-Mia40 as shown in the figure 3C, they may be found in the discarded pellet. Soluble Q97-GFP (~60kD) in this figure is more abundant than that of 6h after induction in the figure 1E in wild-type cells. The authors are asked to explain where the difference is coming from.

At early time points (30-60 min of induction), Q97-GFP remained SDS-soluble, whereas after 6 and more hours of induction, all Q97-GFP was SDS-resistant. However, it should be noted that some aggregates are SDS-soluble and others are not. This is also obvious from the mass spec data shown in our response to point 2 of referee 1.

7) Figure 3C : The aggregation distribution of Q97-GFP in wild-type cells in this figure is different from the figure EV1. In the figure EV1, after 4 h, there are bigger dots of Q97 aggregates in WT than in GAL-Mia40 strains, what is in contrary to the author's interpretation in the figure 3C. The authors should explain what makes the difference. Aggregation mode of Q97 in the presence of GAL-Mia40 in each cell can depend on the expression level of each protein. This might justify the differences presented in the figure 3D. Therefore, the authors are asked to also visualize Mia40 in the microscopy images and provide data that will present levels of Mia40 and Q97 in each cell at the same time, only then the conclusions can be driven.

The authors should comment on the change of the Q25-GFP signal distribution in GAL-Mia40/WT presented in the figure.

As already suggested by the referee, the growth conditions in the Erlenmeyer flasks shown in Fig. 1 and the microfluidics chambers used for Fig. 3 are not identical. This explains why the galactose-dependent kinetics of polyQ induction is slightly different. Importantly, the patterns we observed and the sequence of events are the same under both growth conditions.

8) Figure 3D : The authors quantified the Q97-GFP patterns of aggregates. However, it is unclear what parameters authors used to define which one is small and which one is big. Big dots are visible 6h after induction in WT in the figure 1F, but they were not counted as big dots in the figure 3D (upper graph). The authors should explain how the dots were quantified and why small multiple dots appear 12h after induction in GAL-Mia40 strain.

The aggregates in wild type and GAL-Mia40 cells are actually very different and easy to distinguish. In GAL-Mia40 cells, only one aggregate per cell is formed, which is well defined, bright and round. In wild type cells, several aggregates are present in each cell, they are more patchy and frayed. Actually, these two types of Q97-GFP aggregates were also observed before and are well documented in the literature. However, we are the first to report that their appearance is influenced by mitochondrial proteins.

9) Figure 5B, E : There is less Rnq1 in the soluble fraction of GAL-Mia40 strain compared to that of WT 4,5h after Q97 induction in mass spectrometry data (Fig 5B). However, Rnq1 is more abundant in the soluble fraction of GAL-Mia40 strain than in that of WT (Fig 5E). The authors should explain the discrepancy.

The different aggregation behavior of Rnq1 determines how well the protein is recovered in our samples, digested by trypsin and detected by the mass spectrometer. As shown in Fig. 1, mass spectrometry can be 'blind' for certain states of a protein. Therefore, despite of the accuracy of the instrument and the TMT technology for quantification, proteomics might not be suited to compare the levels of proteins that are present in different states of aggregation. This is why we concluded that there are differences for the Rnq1 protein in both samples, but did not draw conclusions on the nature of these differences. However, these became obvious from the other data shown on Fig. 5.

10) Figure 5E, F : Rnq1 is insoluble in the GAL-Mia40(N), whereas it is soluble in the previous GAL-Mia40 strain. The authors should show whether the GAL-Mia40(N) reproduces the data in the figure 3B, C and D. The authors should examine the Mia40 expression and Q97-GFP aggregation in the GAL-Mia40(N). Additionally, the authors should explain how they exclude the possibility that effect of GAL-Mia40 adaptation doesn't stimulate the secondary effect which are actually responsible for the change of Q97 aggregation. The authors should include such discussion in the manuscript.

Instead of repeating the data with the GAL-Mia40(N) strain, we now show that overexpression of other mitochondrial import components can likewise repress Q97-GFP. We also show that initially not all colonies were able to grow upon polyQ induction (Fig. EV4C) and only few cells were able to escape the growth arrest. However, upon longer time on galactose, cells became polyQ resistant. This nicely recapitulates our

observations for *GAL-Mia40*. It appears likely, that again the effect on the Rnq1 state determines the polyQ toxicity. It should be noted that only the overexpression of specific proteins (Erv1, Tim9, Sam50, Sam37, Hsp104) induces this effect.

11) Figure 6B: the same experiment should be done with *GAL-Mia40+empty* as a control of *GAL-Mia40+Q97*. The quality of the western blot data is not satisfactory and it does not allow to draw any conclusion. If not improved the data and conclusions drawn from it should be removed from the manuscript.

We removed this experiment as suggested by the referee and replaced it by more compelling data, using other approaches. The original data actually was an autoradiography, rather than a Western blot. We hope that the newly added data will be clearer.

12) Figure 6C: in the mass spectrometry data of Table EV1, TOM20, TOM22, TOM70 are also significantly increased in Q97 expressed cells. However the authors focus only on Mia40 as the factor main factor. The authors should check whether any of TOM proteins have the same effect as Mia40. Mia40 upregulation is around 25% due to Q97 expression. The authors show the Mia40 overexpression that significantly exceeding 25% prevents Q97 aggregation. Therefore this is artificial experimental setup. The authors should extend discussion how relevant it is in physiological condition and what could stimulate such big Mia40 response in cells.

We followed the suggestion of the referee and tested whether overexpression of TOM subunits suppresses polyQ toxicity. As shown in novel Figs. 7B and EV4C, they do not.

In addition, we extended the discussion as suggested.

13) Figure 6E: the authors claim that elevated Mia40 is beneficial under stress conditions showing *GAL-Mia40* strain is less reactive to heat stress. HSF1 activation is generally thought to be required for protection from proteotoxicity, so it is questionable whether less HSF1 activation is indeed beneficial to cells. The authors should provide evidence that *GAL-Mia40* strain has less proteotoxicity under heat shock stress than WT. (e.g., PROTEOSTAT staining of WT and *GAL-Mia40* strains after heat shock). Before drawing conclusions, the authors should also include statistical analysis in this figure 6E, and also include the data of *GAL-Mia40+Q97*.

As suggested, we performed an experiment to assess the heat sensitivity of wild type and *GAL-Mia40* cells. As shown in the novel Fig. EV5B, Mia40 overexpression does increase the heat sensitivity, so that cells have problems to grow at 37°C even without expression of Q97-GFP. This points to a potentially protective role of Rnq1 aggregates against high temperature. The analysis of the heat shock response under conditions of Mia40 overexpression will be interesting, but we feel that this will be beyond this study and has to be addressed in more depth in the future.

14) Figure 7B, D, EV4A: MIA40-HA seems to be fully merged with Q97-GFP in human cells. Does this data indicate MIA40 directly interact with Q7-GFP?

We have no experimental evidence for a direct interaction between Q97-GFP and MIA40. In particular, we did not observe an increase in Q97-GFP signal intensity in close proximity to the MIA40 signal. As shown in the example below, we rather see a decrease

in the Q97-GFP signal intensity at MIA40-positive structures. The remaining Q97-GFP signal at MIA40 structures can be attributed to the thickness of the confocal plane, which after processing by structured illumination microscopy (SIM) using the 642 nm excitation laser may reach up to 144 nm. Moreover, light from adjacent confocal planes could contribute to the apparent signal to a small extent.

Fig. 5: The images shown here correspond to figure EV6A (former 4A). Magnification of the merged image illustrates the localization of HA-MIA40 WT (magenta) and Htt-Q97-GFP (green). The arrows indicate the differences in the localization of HA-MIA40 and Htt-Q97-GFP.

15) Figure 7E: The summary figure should be focused on the data presented in the paper. There are not enough evidence presented in the manuscript to make such summary.

We have redrawn the figure. However, it should be emphasized that this is not the summary figure of the data, but a hypothetical model based on our results, that is used to illustrate the Discussion.

16) page 12, last paragraph: the authors state that mitochondrial carrier protein levels are decreased with Q97 expression. To complete the statement, the authors should examine if this is just specific response to carrier proteins.

The carriers were particularly affected, however, also the levels of other mitochondrial proteins are influenced by the expression of Q97-GFP. These changes were moderate in all cases (at least at the 4.5 h time point measured here). Therefore we only mentioned the carrier proteins, because in this group we observed a consistent effect, whereas the influence of Q97-GFP on other protein groups was more heterogeneous.

- minor changes to be addressed/corrected

1) The method of western blotting analysis is missing

We did not describe very basic lab methods, including SDS-PAGE or Western blotting. Even if these methods would be carried out slightly different, it would not make a difference in respect to the conclusions.

2) The method of statistical analysis is missing

We explained this in the respective figure legends.

3) standard deviations/errors of the mean are missing in the figure 2B and C

These are missing because here we show representative results, thus data from single traces.

4) Figure 3B: the lower exposure blot should be added because based on the data, the level of Q25 in GAL-Mia40 seems to be increased when compared to wild-type.

Indeed, the signal of the Q25-GFP protein is considerably stronger than that of the Q97-GFP proteins, even that in the *GAL-Mia40* strain.

5) Figure 4D: indicate what 'M' and 'D' mean in the figure legend.

We added an explanation for mother (M) and daughter (D).

6) Figure 5G: the authors should present the drop dilution experiment data of the WT(N) in the figure.

We decided not to overload the figure with even more data, in particular since the Q97-GFP was already toxic in the previous version. However, this piece of data is actually shown in the novel Fig. 7B, where a new WT version is shown as the upper row, again confirming polyQ toxicity in wild type.

7) Figure 6A: the authors should support data with additional figures of the CHX chase assay showing cytosolic and mitochondrial Mia40 protein levels within the frame of experiment.

Figure 6A showed (and still shows) a drop dilution experiment with different Mia40 versions. Cycloheximide was not used here.

8) Figure 6G: this data should be introduced in more details for non-expert readers.

This graph shows how similar the expression of each gene is in comparison to the genes that encode factors relevant for proteostasis (chaperones and the ubiquitin-proteasome system). This shows that cells coordinate the levels of Mia40 with those of chaperones and other proteostasis factors.

9) Figure 7B, D: the fluorescence intensity of TOM20 and TRAP1 should be increased. They are hardly visible in ΔN -MIA40 expressed cells.

In the revised version, the fluorescence intensity has been increased accordingly.

10) page 3, line 12: use of "Gruber, Hornburg et al., 2018" citation is not appropriate. The authors should differentiate the aggregates association with membranes in yeast and mammals.

We revised the citations accordingly.

11) end of page 4: the authors should specify whether the relation of Mia40 and Rnq1 is based on their result, or an appropriate citation is needed.

This is our result.

12) page 5, first sentence in the result section: the authors should appreciate Susan Lindquist's contribution in the field.

We already had the Duennwald & Lindquist, 2006 paper cited here but now added a further study. Without doubt, Susan Lindquist's contribution to this topic was extremely important and we admire her for her work.

13) page 5, end of first paragraph: the authors should select a better citation. "Duennwald et al., 2006" is based on Q25 and Q103 not Q97.

We do not think that in the context here, the difference of Q97 or Q103 is relevant. Nevertheless, to avoid confusion, we removed the citation as we describe here our own observations.

14) end of page 6: the authors should explain how Mia40 was selected for this study.

We did this by writing: 'the discovery that cytosolic precursors of mitochondrial proteins are toxic was initially made in such a temperature-sensitive Mia40 mutant (Wrobel et al. Nature 2015)'.

15) page 9: "these numbers" -The authors should change it to more precise language.

We changed this as suggested.

16) page 10, end of second paragraph. There is no information to which figure this result section corresponds to. Reference to Figure 5B should be added in an appropriate place.

We corrected this.

17) top of page 12: The authors should provide a citation for the statement of no indication of Mia40 presence in extra-mitochondrial fractions or perform the fractionation experiment supporting this statement.

We did not show these fractionation data since the cytosolic Mia40 variant used in Fig. 6A did anyway not suppress polyQ toxicity.

18) page 14, line 5: "Fig. 6G" should be replaced by "Fig. 6F"

We corrected this.

19) page 15, line 11: "EV4B" should be replaced by "EV4A"

We corrected this.

A**B**
C
Supplemental data referring to Figure 2A.

A: left; B: middle; C: right
non-deconvoluted z stack images (every 2nd layer shown) with
single fluorescence channels and merged fluorescence signals.
scale bar: 5 μm

Thank you for submitting your revised study. The manuscript has now been sent back to the original referees, whose comments are appended below.

As you will see, reviewers #1 and #2 find that their criticisms have been adequately addressed and recommend the study for publication here. However, referee #3 still has few remaining points that I would ask you to address.

In addition, there are few editorial issues concerning the text and the figures that I need you to address before we can officially accept your manuscript.

Referee #1:

This is a revised version of the previously submitted manuscript. The authors responded most of my concerns and questions appropriately, and added substantially new results, so that the manuscript become much stronger. In particular, Fig. 6D and E and Fig. EV4C are quite interesting and raise many new questions, but they will be a subject for future studies. The manuscript thus can contribute a lot to our understanding of the relationship between the mitochondrial protein import and cytosolic quality control, which is a new area of the cell biology studies, and is in a good shape for publication.

Referee #2:

The authors addressed all issues raised on the previous submission. This is a very exciting study of excellent quality. I support publication in its present form.

Referee #3:

The authors have addressed many concerns with additional data or comments. The effort they have made in response to the questions and concerns is highly appreciated. However, some concerns are left not answered, and it is important to address them to match EMBO Journal publication standards.

- Major concerns

- 1) Figure 1E and Figure 3B: To support data interpretation of these two figures, both of them should be supplemented by the SDS-PAGE analysis of the pellet fraction as requested before. This control is important, two reviewers have brought this point up. Otherwise the data interpretation is arbitrary.
- 2) With respect to the original comment 7- Figure 3C and EV1: Authors did not fully answered reviewer's questions, since they did not provide the experiments including visualization of the MIA40 in the microscopic images. However, for the time efficiency of the revision process the authors should include in the paper SDS-PAGE analysis of the polyQ aggregates including 4h and 12h expression and visualize MIA40 expression levels on the blot at the same time.
- 3) With respect to original minor comment 7 - Figure 6A: can the authors comment on the Mia40 protein levels?

-Minor corrections

- 1) The method of statistical analysis is still missing in some figures (Figure 5C-I, Figure 7C,E).
- 2) Figure 2B, C : In this case, the authors should decide if they put a representative result and correct the legend by removing "Mean values of three replicates are shown". Alternatively, they should show the figure matching their legend.
- 3) With respect to original minor comment 4 - Figure 3B: please include the appropriate statement in the text.

Specific points of the referees

Referee #1:

This is a revised version of the previously submitted manuscript. The authors responded most of my concerns and questions appropriately, and added substantially new results, so that the manuscript become much stronger. In particular, Fig. 6D and E and Fig. EV4C are quite interesting and raise many new questions, but they will be a subject for future studies. **The manuscript thus can contribute a lot to our understanding of the relationship between the mitochondrial protein import and cytosolic quality control**, which is a new area of the cell biology studies, and is in a good shape for publication.

We thank referee 1 for her/his very positive evaluation of our study.

Referee #2:

The authors addressed all issues raised on the previous submission. **This is a very exciting study of excellent quality.** I support publication in its present form.

We thank this referee for these very positive statements about the relevance of our observations and the quality of our study.

Referee #3:

The authors have addressed many concerns with additional data or comments. The effort they have made in response to the questions and concerns is highly appreciated. However, some concerns are left not answered, and it is important to address them to match EMBO Journal publication standards.

- Major concerns

1) Figure 1E and Figure 3B: To support data interpretation of these two figures, both of them should be supplemented by the SDS-PAGE analysis of the pellet fraction as requested before. This control is important, two reviewers have brought this point up. Otherwise the data interpretation is arbitrary.

This comment is due to a misunderstanding. In the Western Blots shown in Fig. 1E and 3B, totals of the samples were loaded. Thus, there are no pellet fractions that could be analyzed. We assume that this misunderstanding was caused by our labeling of the Q97-GFP blot where we indicated the soluble and the aggregated species. With soluble, we referred to the SDS-soluble fraction, not the supernatant. To avoid a misunderstanding by our readers, we now replaced 'soluble' by 'monomeric'.

2) With respect to the original comment 7- Figure 3C and EV1: Authors did not fully answered reviewer's questions, since they did not provide the experiments including visualization of the MIA40 in the microscopic images. However, for the time efficiency of the revision process the authors should include in the paper SDS-PAGE analysis of the polyQ aggregates including 4h and 12h expression and visualize MIA40 expression levels on the blot at the same time.

We added now an additional experiment (novel Fig. EV3 B) which shows the levels of Mia40 in the pellet and soluble fractions as suggested. It shows that Mia40 remains largely soluble, even upon overexpression, irrespective of the behavior of Rnq1.

In addition, we show here an experiment in which we assessed the levels of Mia40 in the different strains upon growth on galactose for 18 h. There is no considerable difference in Mia40 levels between strains expressing Q25- and Q97-GFP, in line with the results of the much more accurate proteomics data that are included in the study. Since we felt that these confirmatory data do not strengthen the study but only inflate it, we decided not to include it into the manuscript. However, if the referee or the editor would prefer to have these data included, we would be happy to do so.

Figure for the referee. The levels of Mia40 were assessed by Western blotting and quantified. Three biological replicates are shown. These data confirm the observations of the proteomics data shown in Fig. 5B and 7A, according to which the GAL-Mia40 strain has considerably more Mia40 upon growth on galactose, whereas the expression of Q97-GFP had no strong effect on Mia40 levels.

3) With respect to original minor comment 7 - Figure 6A: can the authors comment on the Mia40 protein levels?

Fig. 6A shows that expression of Q97 leads to a general increase in mitochondrial proteins (see many yellow data points on the right-hand side of the graph). This also includes Mia40. However, this trend is in contrast to what we observed for carriers, most of which are present in reduced amounts. Our interpretation is that the import of these hydrophobic proteins is less efficient, causing an increased degradation of their precursors.

-Minor corrections

1) The method of statistical analysis is still missing in some figures (Figure 5C-I, Figure 7C,E).

As indicated, there are only mean values and standard deviations shown. Statistical analysis was not performed in these experiments.

2) Figure 2B, C : In this case, the authors should decide if they put a representative result and correct the legend by removing "Mean values of three replicates are shown". Alternatively, they should show the figure matching their legend.

We removed the sentence as suggested.

3) With respect to original minor comment 4 - Figure 3B: please include the appropriate statement in the text.

We added a comment to the text and wrote now: "The Mia40-induced suppression was not due to reduced Q97-GFP expression as *GAL-Mia40* cells even contained higher Q97-GFP levels than wild type cells, albeit the levels of Q97-GFP were much lower than those of th

2nd Revision - Editorial Decision

31st May 2021

I am pleased to inform you that your manuscript has been accepted for publication in The EMBO Journal.

Corresponding Author Name: Johannes M Herrmann

Manuscript Number: EMBOJ-2021-107913